# Unary Feedback as Observation: Incentivizing Self-Reflection in Large Language Models via Multi-turn RL

## Abstract

Large Language Models (LLMs) are increasingly deployed as agents that solve problems through multi-turn interaction, receiving feedback and refining their reasoning based on users' feedback. However, existing reinforcement learning with verifiable reward (RLVR) methods train them under a single-turn paradigm. As a result, we discovered that models often fail to explore alternative reasoning paths or reflect on prior mistakes, producing repetitive and unadapted responses to feedback. To address this gap, we propose Unary Feedback as Observation (UFO), a framework that conditions policy updates on minimal unary feedback (e.g., "Let's try again") after incorrect answers. UFO is simple, compatible with existing single-turn RL setups, and incentivizes self-reflection. To further promote efficient and adaptive reasoning, we design reward structures that encourage *minimality* (solving in fewer turns) and *diversity* (exploring alternatives under failure). Experiments show that UFO preserves single-turn performance while improving multi-turn reasoning accuracy by about 14%. Crucially, UFO-trained models also generalize beyond their training domain, transferring effectively to out-of-domain tasks across mathematics, STEM, QA, and general knowledge, showing that UFO teaches models self-reflective reasoning that carry over across domains. Beyond these empirical gains, UFO points toward a broader paradigm for building adaptive reasoning agents: one that scales supervision from static datasets, reduces dependence on costly domain-specific feedback, and lays the foundation for more general, self-improving AI systems in open-ended real-world settings.

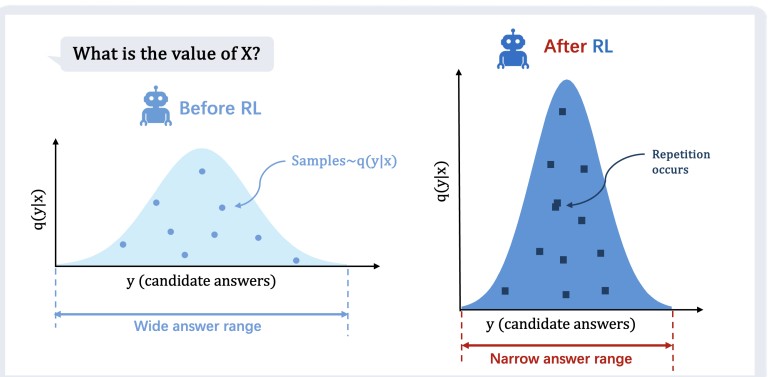

Figure 1: Single-turn RL sharpens the output distribution $q(y|x)$, reducing entropy. During multi-turn interaction, repeated sampling from the narrow distribution leads to high repetition instead of exploration and revision.

## 1 Introduction

Large language and reasoning models (LLMs/LRMs) (DeepSeek-AI, 2025; OpenAI, 2024) have achieved impressive results in domains like maths and coding. Reinforcement learning with verifiable reward (RLVR) (Schulman et al., 2017; Zhou et al., 2024) further enhances their reasoning

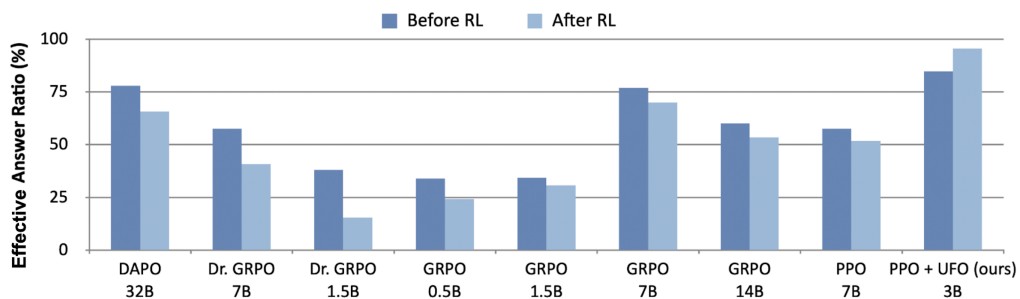

Figure 2: **Comparison of effective answer ratio (%) before and after RL training**. Effective answer ratios drop consistently across different model sizes.

abilities, even enabling models to solve IMO-level math problems (Huang & Yang, 2025). However, real-world applications increasingly require LLMs to act as *agents*, including chatbots, programming assistants, or educational tutors (Xie et al., 2024; Pan et al., 2024). This requires models not only to produce correct outputs, but also to *adapt their reasoning based on users' feedback over multiple turns*. Such settings place higher demands on LLMs and RL methods, as they must support adaptation, exploration, and self-reflection. However, extending single-turn RL training to this broader agentic paradigm remains underexplored.

**A central obstacle is the lack of suitable supervision:** current RLVR is mostly built entirely on single-turn datasets, where models are only trained to output a final answer. We find that in this regime, policies often collapse into single-turn heuristics, showing little incentive to revise or adapt across turns (Figure 1), and this phenomenon persists across RL methods and models scales (Figure 2). Moreover, real-world multi-turn user feedback is costly to collect, leaving few opportunities to train interactive behaviors directly. Existing frameworks therefore rely on automatic feedback such as code interpreter messages (Xie et al., 2024; Wang et al., 2024a) or simulator signals (Shridhar et al., 2021; Zhuang et al., 2025), but these remain narrow in scope and expensive to construct (Cao et al., 2025). This gap motivates our central question: *Can language models be trained to move beyond single-turn heuristics and develop adaptive, self-reflective reasoning behaviors?*

To address this, we propose **Unary Feedback as Observation (UFO)**, a simple framework that transforms static single-turn datasets into multi-turn interaction trajectories by injecting minimal feedback (e.g., "try again"). Unlike prior methods, UFO provides a general mechanism to condition policy updates on failure history. By framing multi-turn reasoning as a Markov Decision Process (MDP) where unary feedback serves as the observation, UFO explicitly trains models to respond to failure, revise prior attempts, and explore alternative reasoning paths.

Experiments show that UFO improves in-domain multi-turn success rates by about 14% without sacrificing single-turn performance. More importantly, UFO-trained models generalize beyond their training domain, transferring effectively to out-of-domain tasks across maths, STEM, QA, and general knowledge. This cross-domain generalization provides direct evidence that UFO instills **adaptive, self-reflective reasoning behaviors**. To further align model behavior with real-world agentic objectives, we introduce two guiding principles: **minimality**, ensuring that an agent can accomplish tasks in as few interactions as possible, and **diversity**, encouraging agents to explore alternative strategies when facing failure. We operationalize these principles via reward decay and answer-repetition penalties, improving both the efficiency and adaptability of multi-turn reasoning.

Our contributions are:

- Identifying the *persistence problem* in single-turn RL, where models fail to adapt their behavior as interactive agents in multi-turn reasoning, and providing a theoretical analysis that explains its low-entropy repetition behavior and establishes the advantage of UFO.

- Proposing **UFO**, a simple multi-turn RL framework that treats unary feedback on static datasets as agent observations. UFO improves in-domain multi-turn reasoning accuracy, and generalizes to multiple out-of-domain tasks, showing its ability to induce **self-reflective reasoning behaviors.**

- Showing that turn-wise reward decay and answer repetition penalty could effectively improve multi-turn reasoning minimality and diversity.

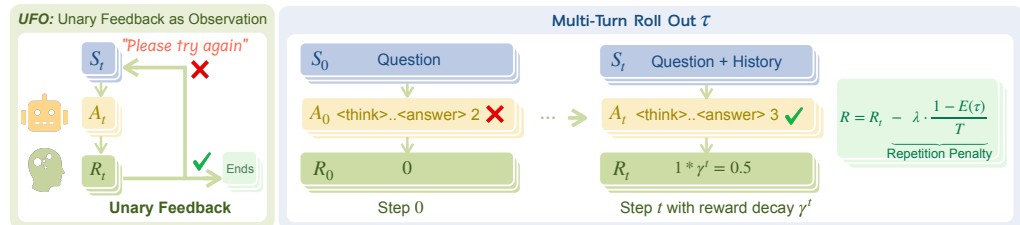

Figure 3: **The UFO framework for multi-turn training.** At each step $t$, the model observes the full interaction history and generates a response. Correct responses receive discounted rewards $\gamma^t$, while incorrect ones receive none. A repetition penalty based on the uniqueness of trajectory $\tau$ is applied after success or when the turn limit is reached.

## 2 FROM SINGLE-TURN TO MULTI-TURN RL

### 2.1 BACKGROUND AND MOTIVATION

**Single-Turn Reinforcement Learning.** Reinforcement Learning (RL) provides a general framework for steering the behavior of large language and reasoning models (LLMs/LRMs). Given a prompt $x \sim \mathcal{D}$, a policy $\pi_\theta$ parameterized by $\theta$ generates a response $y$, which is scored by a reward $R(x, y)$:

$$\max_\theta \ \mathbb{E}_{x\sim\mathcal{D}, \, y\sim\pi_\theta(\cdot|x)}[R(x,y)].$$

Algorithms such as PPO (Schulman et al., 2017) and GRPO (DeepSeek-AI, 2025) apply this objective to static datasets, and have yielded strong improvements in math and code reasoning benchmarks. However, this setup treats reasoning as a *single-turn* decision problem: the model produces one response, receives one reward, and the interaction ends.

**Multi-Turn Extensions.** In contrast, most real-world applications including tutoring systems, coding assistants, or agentic deployments, require *multi-turn* interaction. Here, the model are required to refine its reasoning across turns, incorporating signals from feedback. In some cases, such as programming tasks, dense feedback is available from compilers or unit tests. But in a broader range of open-ended domains—spanning mathematical reasoning, writing, or creative design—feedback is typically sparse. Moreover, obtaining fine-grained human annotations can be prohibitively costly (Pan et al., 2024; Xie et al., 2024). The only generally available guidance is minimal feedback (e.g., "try again"), which indicates failure but not the path to correction. This raises a critical question: can models trained only with single-turn RL generalize to multi-turn reasoning? To answer this, we need to examine how RL-trained models behave when placed in multi-turn settings.

### 2.2 EMPIRICAL EVIDENCE OF COLLAPSE

To assess whether single-turn RL generalizes to interactive settings, we evaluate how RL-trained models behave under minimal feedback. We find that such models are **effective solvers but poor revisers**, consistently failing to incorporate feedback across turns.

Figure 1 illustrates this contrast: a pre-trained model gradually refines its answers, while a single-turn RL model simply repeats its initial output. To quantify this effect, we measure the *effective answer ratio*, i.e., the fraction of distinct attempts a model produces over multiple turns. As shown in Figure 2, RL training consistently reduces this ratio across different algorithms and model scales, indicating a collapse of exploratory behavior. In short, while single-turn RL improves one-shot accuracy, it suppresses diversity in subsequent attempts, leading to degraded multi-turn reasoning.

### 2.3 THEORETICAL ANALYSIS OF COLLAPSE

Why do single-turn RL models repeat mistakes in multi-turn settings? The core reason is that RL training produces a **low-entropy policy**, concentrating probability mass on one response (Cui et al.,

2025; Yue et al., 2025). Formally, the *repetition probability*

$$\text{Rep}(q) := \sum_y q(y \mid x)^2 \tag{1}$$

is bounded below by the distribution's Shannon entropy, $\mathcal{H}(q)$, via

$$\text{Rep}(q) \geq \exp(-\mathcal{H}(q)). \tag{2}$$

implying that decreased entropy inevitably increases repetition (*proof and extensions in Appendix B*).

Avoiding this collapse requires policies that can adapt to interaction history. We distinguish **Parallel Policies** ($\Pi_{par}$), which sample $k$ answers independently from the initial state, from **Sequential Policies** ($\Pi_{seq}$), which condition on prior turns. Since $\Pi_{par} \subseteq \Pi_{seq}$, their success rates obey

$$\max_{\pi \in \Pi_{seq}} \mathbb{E}[\text{Succ@}k] \geq \max_{\pi \in \Pi_{par}} \mathbb{E}[\text{Succ@}k], \tag{3}$$

by Blackwell dominance (Blackwell, 1951); sequential policies can avoid known errors and thus achieve no-worse (often strictly better) performance (*formal statement and proof in Appendix C*).

Together, these results highlight a core limitation: while single-turn RL improves one-shot accuracy, it suppresses LLMs' adaptive revision ability.

## 3 UNARY FEEDBACK FOR MULTI-TURN REASONING

### 3.1 PROBLEM FORMULATION

We formalize multi-turn problem solving on static datasets as a finite-horizon Markov Decision Process (MDP) $(\mathcal{S}, \mathcal{A}, \mathcal{P}, R, T_{\max})$, where $\mathcal{S}$ encodes the question and interaction history, $\mathcal{A}$ is the answer space, $\mathcal{P}$ is the transition function defined by the agent–environment interaction, $R$ provides binary correctness reward, and $T_{\max}$ is the maximum number of interaction steps per episode. At each turn $t$, the agent observes $s_t$ that encodes the original question $q$ and the history of past attempts and feedback:

$$s_t = \texttt{Concat}(q, \{(a_k, f_k)\}_{k=1}^{t-1}), \tag{4}$$

where $a_k$ denotes the $k$-th answer, and $f_k$ is a feedback token returned by the environment. The agent then generates an answer $a_t \sim \pi_\theta(\cdot \mid s_t)$ and receives a scalar reward 1 if $a_t$ is correct, 0 otherwise. The episode terminates once the agent is correct or reaches $T_{\max}$ turns, grounding multi-turn reasoning in a standard RL framework.

### 3.2 UNARY FEEDBACK AS OBSERVATION

To implement the above MDP on static datasets, we introduce **Unary Feedback as Observation (UFO)** (Figure 3), which defines how feedback tokens are represented in the state. UFO restricts feedback to a single negative signal: when an answer $a_k$ is wrong, the agent receives `TryAgain`; when it is correct, the episode terminates with no explicit positive confirmation. Thus, the agent must learn to revise solely from a history of failed attempts.

In practice, the prompt is a natural-language sequence concatenating prior attempts and feedback,

```
Question:  What is the value of ...?
Attempt 1:  [wrong answer]
Feedback:  Try Again.
...
Attempt K: [correct answer]
```

This mechanism transforms static single-turn datasets into multi-turn episodes without structural changes, annotations, or execution environments, enabling multi-turn RL with minimal supervision.

### 3.3 RL WITH UFO

Given the MDP and UFO design, we train the agent with reinforcement learning to acquire revision-aware multi-turn policies. Here, UFO defines the observation structure based on unary feedback,

while PPO supplies the optimization algorithm. Since datasets provide only final-answer accuracy without reasoning traces, supervised finetuning is infeasible, whereas RL can explore diverse strategies under sparse and delayed rewards.

We adopt Proximal Policy Optimization (PPO) (Schulman et al., 2017) with a learned critic for stable value estimation. Each episode produces a trajectory

$$\tau = \{(x_1, a_1, r_1), \dots, (x_T, a_T, r_T)\}, \quad T \leq T_{\max}, \tag{5}$$

where the state $x_t$ encodes prior attempts and unary feedback. Rewards are binary, $r_t \in \{0, 1\}$, reflecting correctness. The objective is to maximize expected return

$$\mathcal{J}^{\mathrm{RL}}(\theta) = \mathbb{E}_{\tau \sim \pi_\theta} \left[ \sum_{t=1}^{T} r_t \right], \tag{6}$$

optimized using the PPO clipped surrogate loss

$$\mathcal{L}^{\mathrm{PPO}}(\theta) = \mathbb{E}_t \left[ \min \left( \frac{\pi_\theta(a_t \mid x_t)}{\pi_{\theta_{\mathrm{old}}}(a_t \mid x_t)} \hat{A}_t, \ \mathrm{clip} \left( \frac{\pi_\theta(a_t \mid x_t)}{\pi_{\theta_{\mathrm{old}}}(a_t \mid x_t)} \hat{A}_t, 1 - \epsilon, 1 + \epsilon \right) \right) \right]. \tag{7}$$

In summary, RL with UFO combines standard PPO with UFO's feedback-driven observation design, enabling policies to turn sparse binary rewards into adaptive multi-turn revision strategies.

### 3.4 Reward Design

Binary correctness signals provide only weak supervision and risk blind trial-and-error. To promote more efficient reasoning, we design trajectory-level rewards with two principles: **minimality**, favoring fewer turns, and **diversity**, discouraging repeated guesses. Reward decay assigns exponentially smaller rewards to later successes:

$$R_t = \begin{cases} \gamma^t, & \text{if } a_t \text{ is correct,} \\ 0, & \text{otherwise,} \end{cases} \tag{8}$$

with $\gamma \in (0, 1)$ controlling the preference for early solutions. Repetition penalty measures answer diversity via the ratio of effective answers $E(\tau)$ to turns $T$:

$$\mathrm{Penalty}(\tau) = \lambda \left( 1 - \frac{E(\tau)}{T} \right). \tag{9}$$

The final trajectory reward combines these terms:

$$R = R_t - \mathrm{Penalty}(\tau), \tag{10}$$

with an additional small penalty $\eta < 0$ for malformed outputs. To summarize, the reward is determined by balancing the reasoning correctness at the last turn, answer diversity and format correctness of model answers across turns.

## 4 Experiments

### 4.1 Setup

**Datasets.** We train on the MATH subset of MetaMathQA (MMQ-Math) (Yu et al., 2024), derived from the MATH competition dataset (Hendrycks et al., 2021), which provides challenging math problems suitable for analyzing reasoning. To evaluate generalization, we use eight benchmarks across four domains: Math: TheoremQA (Chen et al., 2023), GSM8K (Cobbe et al., 2021); STEM: GPQA (Rein et al., 2023), MMLU-STEM; QA: HotPotQA (Yang et al., 2018), ConcurrentQA (Arora et al., 2022); and general: MMLU (Hendrycks et al., 2020), and MMLU-Pro (Wang et al., 2024b). Further dataset details are in Appendix D.

**Training Settings** We use Qwen-2.5-3B-Instruct (Yang et al., 2024) with PPO for 200 optimization steps on A100 GPUs as main training setting. Each batch samples $P=8$ prompts, with $N=16$ rollouts per prompt. During training, we set $T_{\max}$ to 1, 5, and 10 respectively. For the validation phase, $T_{\max}$ is fixed at 5 turns. Policy updates use PPO with GAE parameters $(\gamma, \lambda) = (1.0, 1.0)$, Adam with $\beta = (0.9, 0.999)$, entropy coefficient $10^{-3}$. We apply the same training setup to four additional models: Qwen2.5-1.5B-Instruct (Yang et al., 2024), Qwen2.5-7B-Instruct, LLaMA3.2-1B-Instruct (AI, 2024), and LLaMA3.2-3B-Instruct, to ensure consistent comparison across architectures and scales.

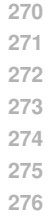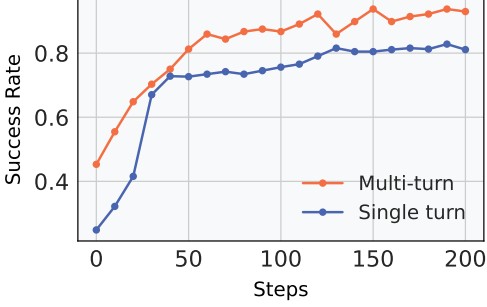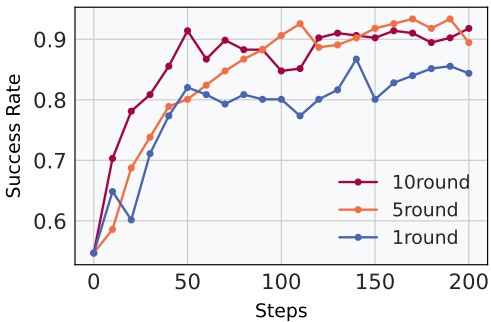

Figure 4: 5-turn UFO significantly outperforms single-turn RL baseline with similar inference cost.

Figure 5: Performance with 5 evaluation turns shows that training with 5 turns performs best, while increasing to 10 offers no clear benefit.

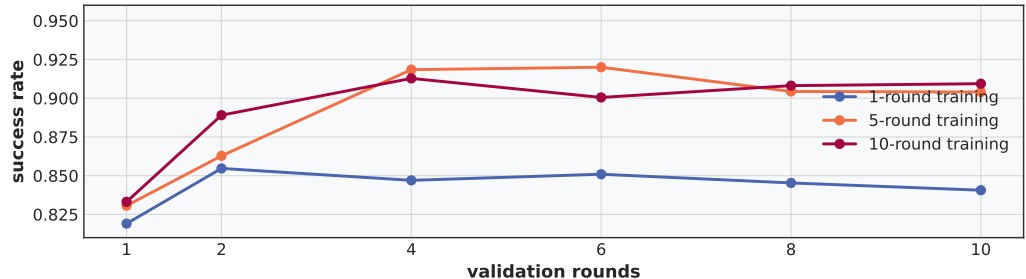

Figure 6: **Validation performance (`Succ@k`) under different training and inference turn settings.** Multi-turn UFO consistently improves success rates across all $k$, including $k = 1$.

**Baselines and Metrics.** We compare our method **UFO** against a single-turn PPO-trained model using parallel sampling. For each problem, the baseline generates $k$ independent responses in parallel and is evaluated using standard `Pass@k` metric. In contrast, our multi-turn model generates responses sequentially with unary feedback after each attempt, and is evaluated using both `Succ@k` and `AvgTurns`. Success is recorded if any of the 5 responses is correct. We also conduct ablation studies with different maximum interaction turns ($T_{\max}$) to further analyze the effect of multi-turn training.

- **Pass@k (Single-turn baseline).** The proportion of problems for which at least one of the k parallel completions is correct.
- **Succ@k (UFO)** This metric measures the percentage of problems solved within a fixed number of interaction turns. Let $\tau_j$ be the number of turns the agent takes to solve problem $q_j$, or $\infty$ if it fails. We have:

$$\text{Succ@k} = \frac{1}{N} \sum_{j=1}^{N} \mathbb{1}[\tau_j \leq k]. \tag{11}$$

We report `Succ@1` for single-turn performance, and `Succ@5/10` to reflect multi-turn capability.
- **Average Number of Turns (UFO)** To evaluate interaction efficiency, we report the average number of turns the agent takes to solve each problem: $\text{AvgTurns} = \frac{1}{N} \sum_{j=1}^{N} T_j$. $T_j$ denotes the number of interactive turns taken for problem $q_j$. This metric reflects how efficiently the agent reaches a solution, accounting for retries and step-wise refinement across multi-turn episodes.

### 4.2 EXPERIMENTAL RESULTS AND FINDINGS

In this section, we present empirical findings that address three central questions in our study of multi-turn reinforcement learning with unary feedback:

1. Section 4.2.1: Does multi-turn RL unlock stronger reasoning than single-turn training?

Table 1: 5-turn success rate across different tasks and training settings.

| Model | Math | | | STEM | | QA | | General | |
|---|---|---|---|---|---|---|---|---|---|
| | MMQ-Math | TheoremQA | GSM8k | GPQA | MMLU-STEM | HotpotQA | ConcurrentQA | MMLU | MMLU-Pro |
| **Qwen2.5-1.5B-Instruct** | | | | | | | | | |
| Base Model w/o RL | 10.9 | 11.7 | 26.6 | 21.9 | 62.5 | 2.4 | 3.1 | 52.3 | 35.2 |
| RL on MMQ-Math | 74.8 | 20.1 | 84.7 | 22.7 | **65.5** | 19.2 | **9.5** | 43.8 | **34.8** |
| +5turn UFO | **83.6** | **26.8** | **88.1** | 27.3 | 64.8 | **22.6** | 9.5 | 60.9 | 34.8 |
| **Qwen2.5-3B-Instruct** | | | | | | | | | |
| Base Model w/o RL | 52.3 | 28.3 | 68.0 | 51.6 | 75.8 | 7.8 | 3.9 | 75.2 | 42.2 |
| RL on MMQ-Math | 79.7 | 32.0 | 93.0 | 50.1 | 77.6 | 19.5 | 12.9 | 66.8 | 48.3 |
| +5turn UFO | **88.5** | **40.8** | **95.3** | 52.3 | 87.5 | 26.6 | 15.2 | **85.2** | **60.9** |
| RL on HotQA | 72.4 | 31.8 | 89.1 | 48.4 | 81.3 | 38.3 | **16.8** | 71.5 | 49.3 |
| +5turn UFO | 72.7 | 29.2 | 85.0 | 57.8 | 88.3 | **44.2** | 16.8 | 76.6 | 48.9 |
| **Qwen2.5-7B-Instruct** | | | | | | | | | |
| Base Model w/o RL | 56.4 | 32.1 | 56.3 | **62.5** | 83.6 | 13.3 | 4.7 | 72.3 | 64.1 |
| RL on MMQ-Math | 85.1 | 33.6 | 95.2 | 50.8 | **84.8** | 26.3 | 14.1 | 73.4 | 52.3 |
| +5turn UFO | **93.0** | **42.1** | **96.8** | 56.9 | 84.8 | 28.6 | 16.4 | 80.5 | 58.8 |
| **Llama3.2-1B-Instruct** | | | | | | | | | |
| Base Model w/o RL* | 2.3 | 2.3 | 1.6 | 1.6 | 4.6 | 0.8 | 0.8 | 3.9 | 2.3 |
| RL on MMQ-Math | 53.9 | 21.1 | 52.3 | 20.3 | 57.0 | 19.5 | 0.8 | 57.8 | **32.8** |
| +5turn UFO | **64.8** | **26.8** | **56.3** | 26.6 | 60.2 | **21.1** | **1.6** | 66.4 | 32.8 |
| **Llama3.2-3B-Instruct** | | | | | | | | | |
| Base Model w/o RL | 50.8 | 20.3 | 48.4 | 47.7 | 77.3 | 29.7 | 6.0 | 65.6 | 49.2 |
| RL on MMQ-Math | 86.7 | 24.2 | 92.2 | 46.9 | 78.1 | **44.5** | 13.3 | 71.1 | 60.9 |
| +5turn UFO | **92.2** | **32.0** | **93.8** | 50.8 | 82.0 | 39.8 | **14.8** | 82.8 | 66.4 |

2. Section 4.2.2: Can models effectively revise their answers from sparse feedback alone?
3. Section 4.2.3: How do reward shaping strategies impact reasoning efficiency and diversity?

We explore each question in the following subsections, with quantitative analyses and ablation studies. Additional qualitative examples and robustness checks are included in the Appendix.

### 4.2.1 MULTI-TURN RL UNLOCKS HIGHER UPPER BOUND OF LLM REASONING

We compare multi-turn RL and a single-turn PPO baseline by measuring `Succ@5` on a validation set, evaluated at 21 checkpoints over 200 training steps. During validation, each agent is allowed up to 5 turns per problem ($k = 5$). As shown in Figure 4, multi-turn training consistently outperforms the single-turn baseline, **achieving 14% higher success rate** with comparable inference cost. Furthermore, we vary training budgets ($T_{\max} = 1, 5, 10$) while validating with $k = 5$ turns. Findings presented in Figure 5 show that both $T_{\max} = 5$ and 10 deliver **more than 6% improvement** over single-turn training, clearly emphasizing the benefits of multi-turn training.

To further validate robustness, we train models with different interaction budgets ($T_{\max} = 1, 5, 10$) and then evaluate them at inference with varying turn limits ($k = 1, 2, 4, 6, 8, 10$). Figure 6 reinforce previous observations, consistently showing the advantage of multi-turn RL training.

### 4.2.2 MULTI-TURN ENABLES LLMS TO REVISE FROM FEEDBACK

Table 1 summarizes results across five models and nine datasets spanning four domains. We highlight three consistent trends:

- **In-domain effectiveness:** Applying 5-turn UFO consistently improves performance over single-turn RL on the same task. [1]
- **Cross-domain generalization:** These gains transfer reliably to Math, STEM, QA, and general knowledge, showing UFO's ability to generalize beyond its training domain.
- **Robustness across scales:** Trends hold across both Qwen2.5 and LLaMA3.2 families (1B–7B), indicating robustness to model scale and architecture.

Together, these findings provide direct evidence for our central claim: training with UFO equips models with **self-reflective reasoning behaviors**. Rather than memorizing single-turn heuristics, models learn to self-reflect, adapt, and transfer reasoning strategies across tasks and domains. The

---

[1] We note that LLaMA3.2-1B shows lower base performance than official numbers, possibly due to format-following issues.

consistency of improvements, spanning five models, two architectures, and nine datasets, demonstrates that these behaviors are not artifacts of a specific setting, but reflect a general capability induced by UFO.

By reusing each prompt across turns, the multi-turn setting effectively converts a single training example into a sequence of interaction signals. Rather than requiring new annotations, this augmentation increases the available data from fixed datasets, turning each problem into a richer trajectory for RL optimization.

We find this data expansion to be the key mechanism enabling models to develop self-reflective behaviors, such as adaptation and strategy revision, under minimal feedback, positioning UFO as a new paradigm for reasoning-focused RL training.

To empirically validate that LRMs can be improved effectively utilizing conversational feedback for revision, we compare 5-turn training scenarios with and without explicit feedback prompts. Results presented in Figure 8 support this hypothesis, **demonstrating an 8% performance improvement when explicit feedback is provided.**

Finally, our robustness analysis in the Figure 8 shows that the effectiveness of this approach is preserved across a range of prompt formulations, underscoring its practical applicability in real-world scenarios.

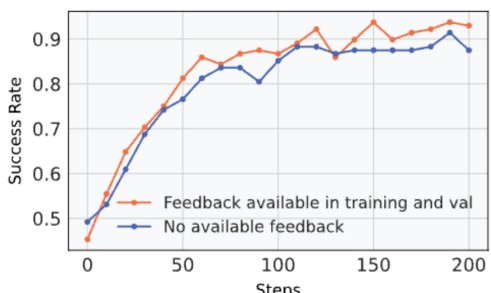

Figure 7: **Effect of feedback prompts in multi-turn training.**

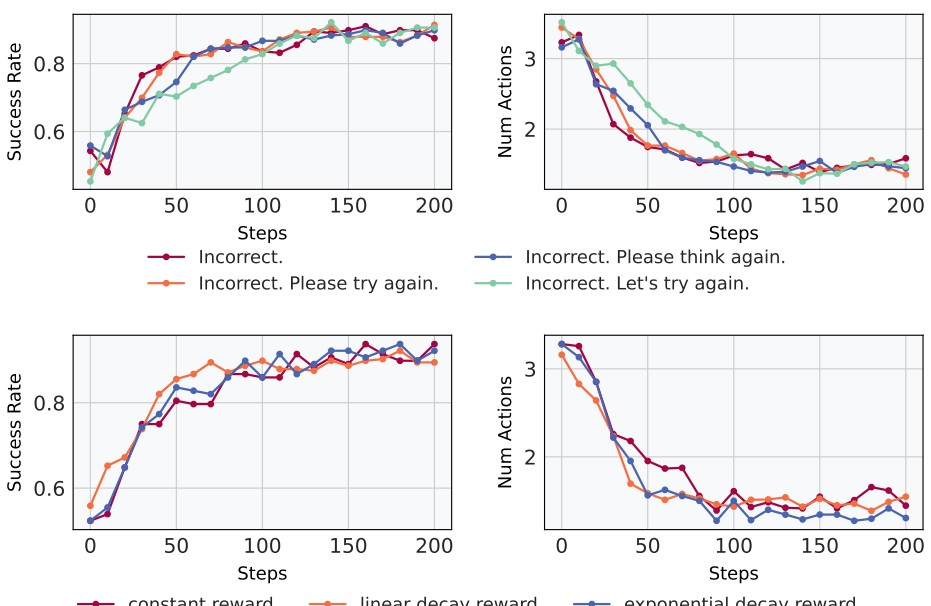

Figure 8: **Analysis of robustness and reward shaping.** Top: Validation under different verbal feedback prompts. Success rates and action counts remain consistent across all variants, demonstrating UFO's robustness to various prompts. Bottom: Comparison of reward shaping strategies. While constant, linear decay, and exponential decay schedules achieve similar success rates (left), exponential decay consistently leads to fewer actions per episode (right), indicating more efficient problem solving with less external supervision.

### 4.2.3 Reward Shaping Encourages Efficient Problem Solving

We investigate how different reward schedules influence the agent's learning behavior, particularly in encouraging early success versus allowing extended exploration. All schedules define a reward $r(n)$ based on the turn index $n$ when the first correct answer is produced, with $n \in \{1, \ldots, T_{\max}\}$.

We define and evaluate three distinct reward schedules. Following the formulas proposed in Section 3.4, we compare three approaches: (1) **Exponential Decay:** $r_{\exp}(n) = \gamma^n$ (with $\gamma = 0.5$), (2) **Linear Decay:** $r_{\lin}(n) = \max(0, 1 - 0.2(n - 1))$, (3) **Constant Reward:** $r_{\const}(n) = 1$. All schedules operate for $n \in 1, \ldots, T_{\max}$. The agent's objective remains to maximize the expected cumulative reward.

Experimental validation (Figure 8) confirms that exponential reward decay notably reduces the mean number of actions by 10%, without sacrificing overall success rates. This reduction in action count suggests that the exponential decay schedule encourages the model to engage in more profound self-reflection and systematic thinking before generating a response.

## 5 Related Work

**Enhancing LLM Reasoning with Test-Time Search and Parameter-Efficient Training.** **Test-time reasoning frameworks** keep model weights frozen yet boost performance: Graph-of-Thought (Besta et al., 2023), Reflexion (Shinn et al., 2023), Monte Carlo Tree Self-Refine (Zhang et al., 2024), Self-Refine (Madaan et al., 2023), CRITIC (Gou et al., 2023) and memory-augmented agents such as POEM (Do et al., 2024) and Larimar (Das et al., 2024) rely on search, self-feedback or episodic memory *without* updating model parameters. **Training-time optimisation methods**, in contrast, adjust the policy itself. RLHF (Christiano et al., 2017; Ouyang et al., 2022) and its low-cost variant RLAIF (Lee et al., 2023) align models to preference data; scalable-oversight *debate* protocols explore alignment with weak judges (Kenton et al., 2024). Lightweight objectives such as Direct Preference Optimisation (DPO) (Rafailov et al., 2023), Parameter-Efficient RLHF (PERLHF) (Sidahmed et al., 2024) and Self-Play Fine-Tuning (SPIN) (Chen et al., 2024) further cut roll-out cost, while hierarchical ArCHer (Zhou et al., 2024) tackles long-horizon credit assignment. Benchmarks like UNO Arena (Qin et al., 2024) expose the strengths and weaknesses of both families in stateful, multi-turn settings.

**Multiturn training for LLMs.** Multiturn training for large language models (LLMs) has been explored across benchmarks, optimization methods, and architectural innovations. Evaluation benchmarks such as LMRL-Gym (Abdulhai et al., 2023) and MT-Eval (Kwan et al., 2024) assess LLMs' abilities to follow instructions and exhibit coherent planning across dialogue turns. On the training side, several works extend RLHF to multi-turn scenarios by optimizing rewards over trajectories, including regression-based value estimation (Gao et al., 2025), hierarchical actor-critic methods (Zhou et al., 2024), and direct preference modeling with trajectory normalization (Shi et al., 2024). Further improvements integrate feedback in mathematical agents (Xiong et al., 2024), while early efforts focus on optimizing full-dialogue preferences (Shani et al., 2024). Additional frameworks such as CollabLLM (Wu et al., 2025) incorporate multi-turn–aware rewards, while RAGEN (Wang et al., 2025) introduces a modular, self-evolving architecture. Beyond RL-based approaches, parameter-efficient fine-tuning methods such as Baize (Xu et al., 2023) demonstrate strong multi-turn capabilities via LoRA adaptation on self-chat data.

## 6 Conclusions

In this work, we identify the *persistence problem* in single-turn RL, where models fail to revise and instead repeat prior mistakes in multi-turn settings. To address this, we propose **Unary Feedback as Observation (UFO)**, a lightweight framework that transforms static single-turn datasets into multi-turn trajectories by augmenting them with minimal unary feedback (e.g., "try again"). This design allows policies to condition on past failures and engage in adaptive self-revision, without requiring costly domain-specific supervision. Empirically, UFO improves multi-turn success rates by $\sim 14\%$ while preserving single-turn quality, and crucially, models trained with UFO generalize across domains—including math, STEM, QA, and general knowledge—demonstrating the emergence of self-reflective reasoning behaviors. Finally, we show that reward designs based on decay and repetition penalties further encourage minimality and diversity, yielding more efficient and adaptive reasoning patterns.

## REPRODUCIBILITY STATEMENT

To facilitate reproducibility, we provide anonymous code in the supplementary materials, which contains all code, training scripts, and configurations necessary to replicate our experimental results. Our theoretical contributions are fully detailed in the Appendix, including assumptions and proofs. Comprehensive dataset descriptions are also included in the supplementary materials. We believe these resources together ensure that both our empirical and theoretical findings can be independently reproduced and verified.

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

## A EXTENDED BACKGROUND OF REINFORCEMENT LEARNING IN LLMS

Reinforcement Learning (RL) enables large language models to improve through interaction and reward feedback. The general RL objective maximizes the expected reward over sampled responses:

$$J(\theta) = \mathbb{E}_{x \sim \mathcal{D},\, y \sim \pi_\theta(\cdot|x)} \left[ R(x, y) \right], \tag{12}$$

where $\pi_\theta$ is the model policy, $x$ is the input prompt, $y$ is the generated output, and $R(x, y)$ is a scalar reward assessing response quality.

A widely adopted method for RL fine-tuning is Proximal Policy Optimization (PPO) (Schulman et al., 2017), which stabilizes training by clipping the likelihood ratio between the new and old policies. The ratio is defined as:

$$\rho_t(\theta) = \frac{\pi_\theta(y_t \mid x_t)}{\pi_{\theta_{\text{old}}}(y_t \mid x_t)}. \tag{13}$$

The PPO objective minimizes over the clipped surrogate advantage:

$$J_{\text{PPO}}(\theta) = \mathbb{E}_t \left[ \min\left( \rho_t A_t, \hat{\rho}_t A_t \right) - \beta D_{\text{KL}} \right], \tag{14}$$

where $\hat{\rho}_t = \text{clip}(\rho_t, 1 - \epsilon, 1 + \epsilon)$, and $A_t$ is the advantage function estimating how much better $y_t$ is than the baseline under prompt $x_t$.

For advantage estimation, Generalized Advantage Estimation (GAE) (Schulman et al., 2015) is often used:

$$A_t^{\text{GAE}(\gamma, \lambda)} = \sum_{l=0}^{\infty} (\gamma \lambda)^l \delta_{t+l}, \quad \text{with} \quad \delta_t = r_t + \gamma V(x_{t+1}) - V(x_t), \tag{15}$$

where $(\gamma, \lambda)$ trade off bias and variance.

More recently, DeepSeekMath (Shao et al., 2024) and DeepSeek-R1 (DeepSeek-AI, 2025) adopts Group Relative Policy Optimization (GRPO), a RL method that samples a set of outputs $\{y_i\}_{i=1}^G$ for each prompt $x$, and optimizes:

$$J_{\text{GRPO}}(\theta) = \mathbb{E}_{x, \{y_i\}} \left[ J_{\text{group}}(\theta) \right], \tag{16}$$

with

$$J_{\text{group}}(\theta) = \frac{1}{G} \sum_{i=1}^G \min\left( \rho_i A_i, \hat{\rho}_i A_i \right) - \beta D_{\text{KL}}, \tag{17}$$

where the advantage $A_i$ is computed using a reward-normalized baseline:

$$A_i = \frac{r_i - \text{mean}(\{r_j\})}{\text{std}(\{r_j\})}. \tag{18}$$

This avoids dependency on value networks and uses rule-based or environment-specific rewards $r_i$, making it well-suited for reasoning tasks where explicit heuristics can guide learning. GRPO has shown to induce emergent multi-step reasoning behavior across domains.

## B DETAILS ON THE THEORETICAL ANALYSIS ON REPETITION PATTERN

We provide a detailed proof on how peaked, low-entropy output distribution from RL training (Cui et al., 2025; Yue et al., 2025) can lead to high repetition in model multi-turn behavior.

**Preliminaries.** Let $q(y \mid x)$ denote the model's output distribution given input $x$. We introduce the following definitions:

**Definition 1** (Repetition Probability). *The repetition probability of $q(y \mid x)$ is defined as:*

$$\text{Rep}(q) := \sum_y q(y \mid x)^2. \tag{19}$$

*This is the probability that two i.i.d. samples from q yield the same answer:* $\Pr[A_i = A_j] = \text{Rep}(q).$

**Definition 2** (Entropy). *The Shannon entropy of $q(y \mid x)$ is:*

$$\mathcal{H}(q) := -\sum_y q(y \mid x) \log q(y \mid x).$$

*Lower entropy corresponds to a more peaked distribution.*

**Definition 3** (Expected Number of Duplicate Pairs). *Given $k$ i.i.d. samples $\{A_1, \ldots, A_k\} \sim q(\cdot \mid x)$, the expected number of duplicate pairs is:*

$$\mathbb{E}[\text{DupPairs}] = \binom{k}{2} \cdot \text{Rep}(q).$$

**Repetition Under Sequential Sampling.**

**Proposition 1.** *Let $A_1, \ldots, A_k \sim q(\cdot \mid x)$ be sampled sequentially. Since single-turn RL does not guarantee any multi-turn capability, we simplify the assumption that the policy is static and does not update based on prior turns, i.e., the answer at each turn $t$ is sampled from the same fixed distribution $q(\cdot|x)$. The probability of generating a duplicate answer is the repetition probability, which is lower-bounded by:*

$$\Pr[A_i = A_j] \geq \exp(-\mathcal{H}(q)),$$

*where $\mathcal{H}(q)$ denotes the Shannon entropy of the base distribution $q(\cdot \mid x)$.*

*Proof.* Let $Y \sim q(\cdot \mid x)$, and define the repetition probability as:

$$\text{Rep}(q) = \mathbb{P}[A_i = A_j] = \sum_y q(y)^2 = \mathbb{E}_{Y \sim q}[q(Y)].$$

By Jensen's inequality applied to the concave function $\log$, we have:

$$\log \mathbb{E}_Y[q(Y)] \geq \mathbb{E}_Y[\log q(Y)] = -\mathcal{H}(q),$$

which implies:

$$\text{Rep}(q) \geq \exp(-\mathcal{H}(q)).$$

$\square$

**Remark 1** (On the Tightness of the Bound). *The lower bound $\text{Rep}(q) \geq \exp(-\mathcal{H}(q))$ is tight in the following cases:*

- *When $q$ is uniform on a support of size $n$, i.e., $q(y) = 1/n$, then $\mathcal{H}(q) = \log n$ and $\text{Rep}(q) = 1/n$, achieving equality.*

- *When $q$ is a delta distribution (i.e., concentrated on one point), then $\mathcal{H}(q) = 0$ and $\text{Rep}(q) = 1$.*

*We assume $\log q(y)$ is only computed where $q(y) > 0$, so the result still holds for distributions with zero-probability points.*

*A tighter bound can be obtained using the Rényi–Shannon inequality:*

$$\text{Rep}(q) = e^{-\mathcal{H}(q) - D_{\text{KL}}(q \| u)},$$

*where $u$ is the uniform distribution on the support of $q$. Our stated bound omits the KL divergence for simplicity and interpretability.*

This demonstrates that a low-entropy model is mathematically guaranteed to have a higher floor for its repetition rate. Thus, for a static agent that does not learn from feedback, repetition is not an accidental bug but a predictable outcome of the low-entropy distributions created by standard RL.

## C  THEORETICAL ANALYSIS: STYLIZED MODEL FOR UNARY FEEDBACK OPTIMIZATION

To better understand why our proposed framework benefits from reward decay and answer diversification, we introduce a minimal stylized model that captures the essence of multi-turn unary feedback (UFO). The model is deliberately simplified with $n$ candidate answers and at most $T$ turns in order to enable tractable analysis, while still retaining the key dynamics of repeated attempts with sparse feedback.

**Setup.** There are $n$ possible answers $A = \{a_1, \ldots, a_n\}$, exactly one of which is correct. Let the prior distribution over the correct answer be $p = (p_1, \ldots, p_n)$ with $p_i > 0$ and $\sum_i p_i = 1$. The agent interacts for at most $T$ turns. At each turn $t$, it selects an answer $a_{i_t}$. If $a_{i_t}$ is correct, the episode ends and the agent receives reward $\gamma^{t-1}$ with $\gamma \in (0, 1]$ modeling decay for delayed success. If $a_{i_t}$ is wrong, the agent observes the unary feedback "Try Again," and continues if $t < T$. We impose *answer diversification*, i.e. no repeats: $i_s \neq i_t$ for $s \neq t$.

**Optimal policy.** At each turn, conditional on previous failures, the posterior over remaining candidates is proportional to their priors. By backward induction, the optimal diversified strategy is *best-first search*: try answers in order of decreasing prior $p_i$ until either success or the turn limit $T$ is reached.

**Success probability and expected return.** Let $p_{(1)} \geq p_{(2)} \geq \cdots \geq p_{(n)}$ denote the sorted priors. Then the probability of success within $T$ turns is

$$\Pr(\text{success by } T) = S_T := \sum_{i=1}^{\min\{T,n\}} p_{(i)}. \tag{20}$$

The corresponding expected return is

$$J_{\text{div}}(T) = \sum_{t=1}^{\min\{T,n\}} \gamma^{t-1} p_{(t)}. \tag{21}$$

In particular $S_T \uparrow 1$ as $T \uparrow n$, so with sufficient turns the agent converges to success with probability 1.

**Convergence rate.** The rate of convergence to 1 is determined by the tail mass $1 - S_T = \sum_{i>T} p_{(i)}$.

- **Uniform prior:** $p_i = 1/n$. Then $S_T = T/n$ and $1 - S_T = 1 - \frac{T}{n}$, i.e. linear convergence.

- **Exponential prior:** $p_{(i)} = c\rho^{i-1}$ with $\rho \in (0, 1)$. Then $1 - S_T = \rho^T$, i.e. geometric convergence.

- **Zipf prior:** $p_{(i)} = c\, i^{-\alpha}$ with $\alpha > 1$. Then $1 - S_T = \Theta(T^{1-\alpha})$, i.e. polynomial convergence.

Thus, the structure of the prior distribution dictates how quickly UFO drives success probability toward 1 as turns increase.

**Comparison to i.i.d. sampling.** If the agent ignores history and samples i.i.d. from some fixed distribution $q = (q_1, \ldots, q_n)$ at each turn, the success probability is

$$\Pr_{\text{iid}}(\text{success by } T) = 1 - \sum_{i=1}^{n} p_i (1 - q_i)^T. \tag{22}$$

This quantity is strictly smaller than $S_T$ in equation 20 whenever $T < n$ and the prior is non-uniform, since repeats waste turns on known-wrong answers. Moreover, with decay $\gamma < 1$, the expected return $J_{\text{iid}}(T; q)$ is dominated by $J_{\text{div}}(T)$ due to the rearrangement inequality: pairing higher-probability answers with earlier attempts maximizes discounted value.

**Implications.** This stylized analysis shows that diversification together with reward decay strictly improves both success probability and expected return relative to i.i.d. sampling. Failure probability decays at a rate governed by the prior tail—linear, geometric, or polynomial depending on prior shape. Crucially, diversification eliminates repeated errors, ensuring every failure eventually yields positive signal, and reward decay biases the agent toward placing its best guesses earlier. Together, these mechanisms explain why UFO training can push multi-turn reasoning toward the theoretical limit of correctness.

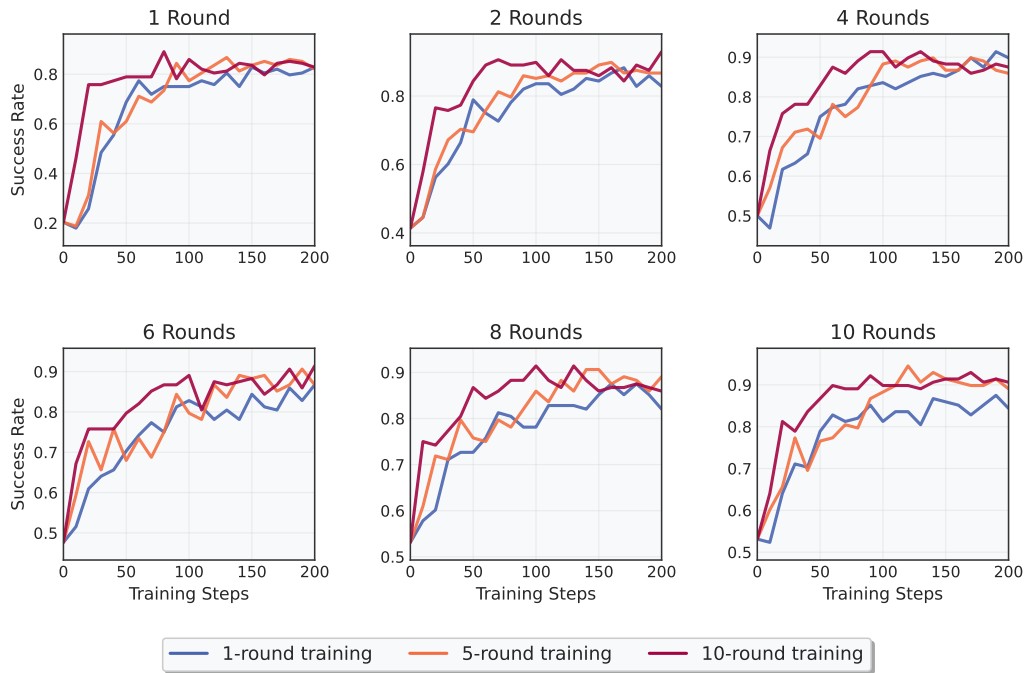

Figure 9: **Performance across different evaluation round settings.** Each subplot shows the success rate evaluated at $r$ rounds. While all methods perform similarly under 1-round evaluation, models trained with multi-round feedback (UFO) generalize significantly better to longer evaluation horizons.

## D    DATASET DETAILS

We evaluate the generalization of UFO on eight widely-used benchmarks spanning mathematics, science, and QA:

- **TheoremQA** (Chen et al., 2023): formal mathematics, theorem statements and proofs.
- **GSM8K** (Cobbe et al., 2021): grade-school arithmetic reasoning.
- **GPQA** (Rein et al., 2023): graduate-level physics problems.
- **MMLU-STEM** (Hendrycks et al., 2020): 15 scientific and technical subjects from MMLU.
- **HotPotQA** (Yang et al., 2018): multi-hop factual reasoning over Wikipedia.
- **ConcurrentQA** (Arora et al., 2022): temporal and causal reasoning across events.
- **MMLU** (Hendrycks et al., 2020): 57 subjects covering broad general knowledge.
- **MMLU-Pro** (Wang et al., 2024b): expert-level extension of MMLU.

## E    DETAILED EVALUATION UNDER MULTI-ROUND SETTINGS

We illustrate a detailed analysis of how multi-round training improves generalization on long-horizon interactive reasoning. Figure 9 provides a comprehensive view of validation performance across all checkpoints, comparing models trained under 1-round, 5-round, and 10-round settings. Each curve represents evaluation success rates under a fixed number of evaluation rounds.

We observe that under 1-round evaluation (top-left), all training strategies achieve similar performance, suggesting that even single-turn training can suffice in this limited setting. However, as evaluation round count increases, the gap between single-round training and multi-round training becomes increasingly significant. In particular, models trained with 10-round UFO feedback consistently outperform the others under 6, 8, and 10-round evaluation, demonstrating more stable and generalizable behavior across turns.

These results support our core hypothesis: **unary feedback, when used as structured observation during training, enables better long-horizon generalization.** In contrast, models trained only with single-round interactions struggle to adapt to multi-turn dynamics, leading to degraded performance as the task horizon increases.

# F  PROMPT SETTINGS

## F.1  PROBLEM SOLVING MODEL PROMPT

We adopt a simple and structured prompt format for mathematical problem solving, following prior designs from Shao et al. (2024); Yang et al. (2024), with an extension to support multi-turn interactions. A key element of our prompt is the explicit `<think>` and `<answer>` separation, paired with an action budget (Y) and max length (Z). This guides the model to reason step-by-step while planning within a fixed turn horizon, improving controllability and alignment in multi-turn settings. As shown in Box 1, we present the prompt template used during training and evaluation.

---

**Box 1: Model Prompt Template**

```
<|im_start|>system
{prompt}
You're a helpful assistant.
<|im_end|>
<|im_start|>user
{prompt}
You are solving Math problems.
Turn X:
State:
(Question)
You have Y actions left.  Always output:  <think> [Your
thoughts] </think> <answer> [your answer] </answer> with
no extra text.  Strictly follow this format.  Max response
length:  Z words (tokens).
<|im_end|>
<|im_start|>assistant
...  (This conversation pattern repeats for up to K turns)
<|im_end|>
```

---

## F.2  FEEDBACK PROVIDER MODEL PROMPT

We also present the prompt for the feedback provider that gives the problem-solving model more detailed feedback as follows.

---

**Box 2: Tutor Prompt Template**

```
<|im_start|>system
You are a helpful math tutor.
<|im_end|>
<|im_start|>user
Problem:  {question}
Student's answer:  {wrong_answer}
This answer is incorrect.  Give a brief, encouraging hint
(1--2 sentences) that helps the student reconsider their
approach without revealing the correct answer.  Focus on
guiding them to check their work or think about the problem
differently.
Response format:  Just the hint, no extra formatting.
<|im_end|>
<|im_start|>assistant
```

---

```
...
<|im_end|>
```

## F.3 THEOREMQA PROMPT FORMAT

The TheoremQA environment follows a similar prompting structure as the MMQ-Math, with an additional image token placed at the beginning of the question when a picture is present. This enables compatibility with multimodal pipelines, where the image is processed separately while the text prompt includes a placeholder token to signal its presence.

As shown in Box 3, the image token `<image>` is placed on a new line above the question if an image is available.

---

**Box 3: TheoremQA Prompt Template**

```
<|im_start|>system
You're a helpful assistant.
<|im_end|>
<|im_start|>user
{prompt}
You are solving Math problems.
Turn X:
State:
<image>
+ (Question)
You have Y actions left.  Always output:  <think>[Your
thoughts]</think><answer>[your answer]</answer> with no extra
text.  Strictly follow this format.  Max response length:  Z
words (tokens).
<|im_end|>
<|im_start|>assistant
...
<|im_end|>
```

---

## G MODEL EVALUATION DETAILS

We present the model used to evaluate answer repetition in Table 2.

## H CASE ANALYSES

We investigate the impact of multi-turn reinforcement learning (RL) on large language models (LLMs) through a series of curated examples across distinct training stages. These case studies (shown in Boxes 4–7) highlight the evolving dynamics of exploration, convergence, and reasoning quality throughout training.

### H.1 CASE 1: PRE-TRAINING BEHAVIOR

Before any reinforcement learning, we observe the model's default multi-turn reasoning behavior in a symbolic pattern-matching task (Box 4). The model is asked to recover a missing variable $X$ from a repeating alphabetic value pattern, given partial information and a constraint on the sum of values in a specific word. In Turn 1, the model identifies the relevant positions in the word "numeric" and proposes an initial guess for $X$. As feedback indicates the answer is incorrect, the model progressively refines its understanding: it attempts to align characters in the input word with their positions in the pattern and adjusts its value for $X$.

Despite making several wrong guesses, the model demonstrates **adaptive behavior across turns**: it updates its assumptions, introduces new hypotheses, and makes meaningful structural progress (e.g., recognizing the 8-length cycle). However, it ultimately fails to reach the correct solution

Table 2: Hugging Face model names used in the unique answer ratio evaluation.

| Method | Model (Hugging Face name) |
|---|---|
| DAPO | Qwen/Qwen2.5-32B |
| | BytedTsinghua-SIA/DAPO-Qwen-32B |
| Dr. GRPO | Qwen/Qwen2.5-Math-7B |
| | sail/Qwen2.5-Math-7B-Oat-Zero |
| | Qwen/Qwen2.5-Math-1.5B |
| | sail/Qwen2.5-Math-1.5B-Oat-Zero |
| GRPO | Qwen/Qwen2.5-0.5B |
| | hkust-nlp/Qwen-2.5-0.5B-SimpleRL-Zoo |
| | Qwen/Qwen2.5-1.5B |
| | hkust-nlp/Qwen-2.5-1.5B-SimpleRL-Zoo |
| | Qwen/Qwen2.5-7B |
| | hkust-nlp/Qwen-2.5-7B-SimpleRL-Zoo |
| | Qwen/Qwen2.5-14B |
| | hkust-nlp/Qwen-2.5-14B-SimpleRL-Zoo |
| PPO | Qwen/Qwen2.5-Math-7B |
| | RLHFlow/Qwen2.5-7B-PPO-Zero |
| | Qwen/Qwen2.5-3B-Instruct |
| | LichengLiu03/Qwen2.5-3B-UFO |

within the available steps. The case shows that **pretrained models already possess multi-step reflective capabilities and can utilize external feedback to revise their reasoning**, even without explicit training for multi-turn alignment. It suggest that reinforcement learning has the potential to further stabilize and guide emergent reasoning process toward convergence.

### H.2 CASE 2: POST SINGLE-TURN RL

After reinforcement learning with single-step reward feedback, the model demonstrates drastically different behavior from its pretrained counterpart (Box 5). When tasked with identifying the variable $X$ in a cyclic pattern-based word problem, the model immediately commits to a single interpretation. In Turn 1, it attempts a symbolic derivation by aligning the letter values of "numeric" with a fixed periodic pattern, then solving $X - 3 = -1$. However, this derivation mistakenly assumes that the letters in "numeric" correspond to the first 7 elements of the pattern without justifying the mapping. More notably, this exact sequence of logic and answer is **repeated identically** in Turns 2 through 5.

**The case reveals that single-turn RL induces brittle, overconfident behavior**: once the model settles on a trajectory during initial inference, it does not reconsider alternative hypotheses or respond meaningfully to corrective feedback. The reward optimization has led to collapse in exploration, as each turn simply replays the same incorrect reasoning with no adaptation. In contrast to the pretraining stage, where the model at least attempts different strategies, this behavior illustrates a major drawback of single-step reward supervision: it teaches the model what to say once, but not how to revise when it's wrong.

### H.3 CASE 3: SUCCESS ADAPTATION TO FEEDBACK THROUGH MULTI-TURN RL WITH UFO

This example illustrates the effectiveness of multi-turn reinforcement learning (Box 6). The model is prompted to determine the sum of all positive integers $n$ for which $\frac{n+18}{n}$ is an integer. In Turn 1, it begins by simplifying the expression to $1 + \frac{18}{n}$, and attempts a partial answer without listing all divisors. Upon receiving feedback, the model updates its understanding in Turn 2 by enumerating all positive divisors of 18. By Turn 3, it completes the reasoning process by summing those divisors correctly, arriving at the correct final answer of 39.

The case shows a successful case of multi-turn self-correction, where the model refines its reasoning incrementally in response to feedback. Each turn builds on the previous one: the model first identifies the mathematical form, then retrieves the correct domain knowledge (divisors), and finally executes a complete and valid computation. Unlike single-turn RL, **the multi-turn reward struc-**

Table 3: Comparison of multi-turn reasoning behaviors across training stages.

| Case | Stage | Exploration | Convergence | Reasoning Quality | Failure Mode |
|------|-------|-------------|-------------|-------------------|--------------|
| 1 | Pre-RL | High | No | Incomplete | Early guessing |
| 2 | Single-turn RL | None | No | Repetitive | Overfitting |
| 3 | Multi-turn RL | Moderate | Yes | Correct | aligned |
| 4 | Multi-turn RL | Moderate | Yes | Incorrect | Reasoning drift |

**ture encourages adaptive reasoning and gradual convergence.** The example shows that properly trained multi-turn agents can both reason correctly and update their behavior mid-trajectory in pursuit of a verifiable answer.

### H.4 CASE 4: REASONING DRIFT OF MULTI-TURN RL WITH UFO

We examine a failure trajectory under multi-turn reinforcement learning where the model arrives at the correct final answer through incorrect or misaligned reasoning. As shown in Box 7, the task is to find the value of $X$ such that the matrix $\begin{pmatrix} X & 3 \\ -1 & 7 \end{pmatrix}$ has an inverse consistent with the given answer "17." In the initial turn, the model misinterprets the problem structure by assuming that the matrix inverse itself is the scalar 17, and raises a structural objection. In the next turn, it shifts to a more plausible interpretation where 17 is assumed to be the determinant. This leads to the formulation of the correct equation $7X + 3 = 17$, though the value of $X$ is still miscalculated at that stage.

By the third turn, the model correctly solves for $X = 2$, but then presents a flawed expression for the inverse matrix, misapplying the standard inversion formula for $2 \times 2$ matrices. Despite producing the right numerical solution, the trajectory contains conceptual errors related to matrix algebra, including an incorrect cofactor layout. This case highlights a form of reasoning drift: **while the final output aligns with the ground truth, the intermediate steps do not constitute a logically valid derivation.** Such behavior reveals a potential failure of reward design in multi-turn RL, where correctness of the answer is reinforced even in the absence of faithful reasoning.

To summarize (as in Table 3), multi-turn RL with UFO improves convergence and stabilizes reasoning, but naive reward design can still lead to misalignment between thought and outcome. It highlights the need for evaluation frameworks to go beyond final answer correctness and explicitly assess reasoning integrity throughout the trajectory.

---

**Box 4: Multi-Turn Rollout before training**

**Turn 1:**
**State:** The letters of the alphabet are given numeric values based on the two conditions below.
• Only the numeric values of $-2$, $-1$, 0, 1 and 2 are used.
• Starting with A and going through Z, a numeric value is assigned to each letter according to the following pattern:

$$1, 2, X, 0, -1, -2, -1, 0, 1, 2, 1, 0, -1, -2, -1, 0, \ldots$$

Two complete cycles of the pattern are shown above. The letter A has a value of 1, B has a value of 2, F has a value of $-2$ and Z has a value of 2. The sum of the numeric values of the letters in the word "numeric" is -1. What is the value of unknown variable X? You have 10 actions left. Always output: `<think>` [Your thoughts] `<think>` `<answer>` [your answer] `<answer>` with no extra text. Strictly follow this format. Max response length: 100 words (tokens).
**Output:**
`<think>`First, let's identify the positions of each letter in the word "numeric" based on the pattern provided. We know the values of A, B, F, and Z. We need to determine the value of X and then find the values of N, M, E, and C to sum up to -1.`</think>`
`<answer>`X = 2`</answer>`
Reward: 0.0

---

**Turn 2:**
**State:** Incorrect. Please try agin. You have 3 actions left. Always output: `<think>` [Your thoughts] `<think>` `<answer>` [your answer] `<answer>` with no extra text. Strictly follow this format. Max response length: 100 words (tokens).
**Output:**
`<think>`Given the incorrect answer, let's re-examine the pattern and the values assigned to each letter. The word "numeric" has the letters N, U, M, B, E, I, C. From the provided pattern, we know the values of A, B, F, and Z. To find the value of X, we need to identify the value for N and then determine the pattern for subsequent letters.`</think>`
`<answer>`X = 0`</answer>`
Reward: 0.0

**Turn 3:**
**State:** Incorrect. Please try agin. You have 2 actions left. Always output: `<think>` [Your thoughts] `<think>` `<answer>` [your answer] `<answer>` with no extra text. Strictly follow this format. Max response length: 100 words (tokens).
**Output:**
`<think>`Given the incorrect answer, let's re-evaluate the pattern and the values assigned to each letter in the word "numeric". The values for A, B, F, and Z are known. The pattern seems to repeat every 8 letters, and the values cycle through 1, 2, X, 0, -1, -2, -1, 0. We need to determine the value of X such that the sum of the values of the letters in "numeric" equals -1.`</think>`
`<answer>`X = -1`</answer>`
Reward: 0.0

**Turn 4:**
**State:** Incorrect. Please try agin. You have 1 actions left. Always output: `<think>` [Your thoughts] `<think>` `<answer>` [your answer] `<answer>` with no extra text. Strictly follow this format. Max response length: 100 words (tokens).
**Output:**
`<think>`Given the incorrect answer, let's re-examine the pattern and the values assigned to each letter in the word "numeric". The values for A, B, F, and Z are known, and the pattern seems to repeat every 8 letters with values 1, 2, X, 0, -1, -2, -1, 0. To find the value of X such that the sum of the values of the letters in "numeric" equals -1, we need to assign values to N, U, M, B, E, I, C and ensure their sum is -1`</think>`
`<answer>`X = 1`</answer>`
Reward: 0.0

**Turn 5:**
**State:** Incorrect. Please try agin. You have 0 actions left. Always output: `<think>` [Your thoughts] `<think>` `<answer>` [your answer] `<answer>` with no extra text. Strictly follow this format. Max response length: 100 words (tokens).
**Output:**
`<think>`Given the incorrect answer, let's re-examine the pattern and the values assigned to each letter in the word "numeric". The values for A, B, F, and Z are known, and the pattern seems to repeat every 8 letters with values 1, 2, X, 0, -1, -2, -1, 0. To find the value of X such that the sum of the values of the letters in "numeric" equals -1, we need to assign values to N, U, M, B, E, I, C and ensure their sum is -1.`</think>`
`<answer>`X = -1`</answer>`
Reward: 0.0

---

**Box 5: Model rollout after single-turn RL training**

**Turn 1:**
**State:** The letters of the alphabet are given numeric values based on the two conditions below.
• Only the numeric values of −2, −1, 0, 1 and 2 are used.

• Starting with A and going through Z, a numeric value is assigned to each letter according to the following pattern:

$$1, 2, X, 0, -1, -2, -1, 0, 1, 2, 1, 0, -1, -2, -1, 0, \ldots$$

Two complete cycles of the pattern are shown above. The letter A has a value of 1, B has a value of 2, F has a value of $-2$ and Z has a value of 2. The sum of the numeric values of the letters in the word "numeric" is -1. What is the value of unknown variable X? You have 10 actions left. Always output: `<think>` [Your thoughts] `<think>` `<answer>` [your answer] `<answer>` with no extra text. Strictly follow this format. Max response length: 100 words (tokens).

**Output:**

`<think>`For the word "numeric", the values of the letters are as follows: N has a value of 1, U has a value of 2, M has a value of 0, E has a value of -1, R has a value of -2, I has a value of -1, C has a value of 1. The sum of these values is $1 + 2 + 0 - 1 - 2 - 1 + 1 = 0$, which contradicts the given sum of -1. However, we can re-evaluate the pattern and the values. The pattern repeats every 16 letters, and the values repeat as well. The word "numeric" consists of 7 letters, so we need to find the sum of the first 7 values in the pattern: $1, 2, X, 0, -1, -2, -1$. We know the sum of these values is -1. So, we have $1 + 2 + X + 0 - 1 - 2 - 1 = -1$, which simplifies to $X - 3 = -1$, and thus $X = 2$.`</think>`
`<answer>`2`</answer>`
Reward: 0.0

**Turn 2:**
**State:** Incorrect. Please try agin. You have 3 actions left. Always output: `<think>` [Your thoughts] `<think>` `<answer>` [your answer] `<answer>` with no extra text. Strictly follow this format. Max response length: 100 words (tokens).
**Output:**
`<think>`For the word "numeric", the values of the letters are as follows: N has a value of 1, U has a value of 2, M has a value of 0, E has a value of -1, R has a value of -2, I has a value of -1, C has a value of 1. The sum of these values is $1 + 2 + 0 - 1 - 2 - 1 + 1 = 0$, which contradicts the given sum of -1. The pattern repeats every 16 letters, and the values repeat as $1, 2, X, 0, -1, -2, -1, 0, 1, 2, 1, 0, -1, -2, -1, 0$. For the word "numeric", the values are $1, 2, X, 0, -1, -2, -1$. The sum of these values is $1 + 2 + X + 0 - 1 - 2 - 1 = -1$, which simplifies to $X - 3 = -1$, and thus $X = 2$.`</think>`
`<answer>`2`</answer>`
Reward: 0.0

**Turn 3:**
**State:** Incorrect. Please try agin. You have 2 actions left. Always output: `<think>` [Your thoughts] `<think>` `<answer>` [your answer] `<answer>` with no extra text. Strictly follow this format. Max response length: 100 words (tokens).
**Output:**
`<think>`For the word "numeric", the values of the letters are as follows: N has a value of 1, U has a value of 2, M has a value of 0, E has a value of -1, R has a value of -2, I has a value of -1, C has a value of 1. The sum of these values is $1 + 2 + 0 - 1 - 2 - 1 + 1 = 0$, which contradicts the given sum of -1. The pattern repeats every 16 letters, and the values repeat as $1, 2, X, 0, -1, -2, -1, 0, 1, 2, 1, 0, -1, -2, -1, 0$. For the word "numeric", the values are $1, 2, X, 0, -1, -2, -1$. The sum of these values is $1 + 2 + X + 0 - 1 - 2 - 1 = -1$, which simplifies to $X - 3 = -1$, and thus $X = 2$.`</think>`
`<answer>`2`</answer>`
Reward: 0.0

**Turn 4:**
**State:** Incorrect. Please try agin. You have 1 actions left. Always output: `<think>` [Your thoughts] `<think>` `<answer>` [your answer] `<answer>` with no extra text. Strictly follow this format. Max response length: 100 words (tokens).
**Output:**
`<think>`For the word "numeric", the values of the letters are as follows: N has a value of 1, U has a value of 2, M has a value of 0, E has a value of -1, R has a value of -2, I has a value of -1, C has a value of 1. The sum of these values is $1 + 2 + 0 - 1 - 2 - 1 + 1 = 0$, which contradicts the given sum of -1. The pattern repeats every 16 letters, and the values repeat

as $1, 2, X, 0, -1, -2, -1, 0, 1, 2, 1, 0, -1, -2, -1, 0$. For the word "numeric", the values are $1, 2, X, 0, -1, -2, -1$. The sum of these values is $1 + 2 + X + 0 - 1 - 2 - 1 = -1$, which simplifies to $X - 3 = -1$, and thus $X = 2$.</think>
<answer>2</answer>
Reward: 0.0

**Turn 5:**

**State:** Incorrect. Please try agin. You have 0 actions left. Always output: <think> [Your thoughts] <think> <answer> [your answer] <answer> with no extra text. Strictly follow this format. Max response length: 100 words (tokens).

**Output:**
<think>For the word "numeric", the values of the letters are as follows: N has a value of 1, U has a value of 2, M has a value of 0, E has a value of -1, R has a value of -2, I has a value of -1, C has a value of 1. The sum of these values is $1 + 2 + 0 - 1 - 2 - 1 + 1 = 0$, which contradicts the given sum of -1. The pattern repeats every 16 letters, and the values repeat as $1, 2, X, 0, -1, -2, -1, 0, 1, 2, 1, 0, -1, -2, -1, 0$. For the word "numeric", the values are $1, 2, X, 0, -1, -2, -1$. The sum of these values is $1 + 2 + X + 0 - 1 - 2 - 1 = -1$, which simplifies to $X - 3 = -1$, and thus $X = 2$.</think>
<answer>2</answer>
Reward: 0.0

---

**Box 6: Model rollout after multi-turn RL training with UFO (Success Case)**

**Turn 1:**
**State:**
Determine the sum of all positive integer values of $n$ such that $\frac{n+18}{n}$ is an integer.
You have 10 actions left. Always output: <think> [Your thoughts] </think> <answer> [your answer] </answer> with no extra text. Strictly follow this format. Max response length: 100 words.
**Output:**
<think>First, I need to understand when $\frac{n+18}{n}$ is an integer. This fraction can be simplified to $1 + \frac{18}{n}$. For this to be an integer, $n$ must be a divisor of 18.</think>
<answer>18 and 1 (since $1 + \frac{18}{18} = 2$ and $1 + \frac{18}{1} = 19$ are integers)</answer>
Reward: 0.0

**Turn 2:**
**State:**
Incorrect. Please think again.
You have 9 actions left.
**Output:**
<think>Given $\frac{n+18}{n}$ is an integer, we have $1 + \frac{18}{n}$ as an integer. This means $n$ must be a divisor of 18. The positive divisors of 18 are 1, 2, 3, 6, 9, and 18. I need to verify which of these values make the expression an integer.</think>
<answer>1, 2, 3, 6, 9, 18</answer>
Reward: 0.0

**Turn 3:**
**State:**
Incorrect. Please think again.
You have 8 actions left.
**Output:**
<think>Upon reviewing, I realize the divisors of 18 are 1, 2, 3, 6, 9, and 18. For each of these values, $1 + \frac{18}{n}$ is indeed an integer. Therefore, the sum of all positive integer values of $n$ is $1 + 2 + 3 + 6 + 9 + 18 = 39$.</think>
<answer>39</answer>
Reward: 0.25

---

**Box 7: Model rollout after multi-turn RL training with UFO (Failure Case)**

**Turn 1:**
**State:**
Find the inverse of the matrix

$$\begin{pmatrix} X & 3 \\ -1 & 7 \end{pmatrix}.$$

If the inverse does not exist, then enter the zero matrix. The answer is 17. What is the value of unknown variable $X$?
You have 10 actions left. Always output: `<think>` [Your thoughts] `</think>` `<answer>` [your answer] `</answer>` with no extra text. Strictly follow this format. Max response length: 100 words.
**Output:**
`<think>`Since the inverse of a matrix is given as 17, and the matrix provided is 2x2, the inverse should be a 2x2 matrix. However, the given inverse of 17 does not align with the structure of a 2x2 matrix. The problem might have a mistake in the provided inverse value or the matrix structure.`</think>`
`<answer>`There is an inconsistency in the problem statement. The given inverse of 17 does not match the structure of a 2x2 matrix inverse.`</answer>`
Reward: 0.0

**Turn 2:**
**State:**
Incorrect. Please think again.
You have 9 actions left.
**Output:**
`<think>`Given the inverse of the matrix is provided as 17, which does not fit the expected format of a 2x2 matrix inverse (which should be a 2x2 matrix), it seems there might be a misunderstanding. However, since the problem statement specifies the inverse as 17, and considering the structure of the matrix, it implies the inverse is a scalar value. For a 2x2 matrix $\begin{pmatrix} a & b \\ c & d \end{pmatrix}$, the inverse is $\frac{1}{ad-bc}\begin{pmatrix} d & -b \\ -c & a \end{pmatrix}$. If the inverse is simply 17, it suggests $ad - bc = 17$. Given the matrix $\begin{pmatrix} X & 3 \\ -1 & 7 \end{pmatrix}$, $ad - bc = X \cdot 7 - 3 \cdot (-1) = 7X + 3$. Thus, $7X + 3 = 17.$`</think>`
`<answer>`7`</answer>`
Reward: 0.0

**Turn 3:**
**State:**
Incorrect. Please think again.
You have 8 actions left.
**Output:**
`<think>`Given the matrix $\begin{pmatrix} X & 3 \\ -1 & 7 \end{pmatrix}$, the condition for the inverse to be a scalar 17 is $7X + 3 = 17$. Solving for $X$ gives $7X = 14$, thus $X = 2$. The inverse of the matrix should be $\frac{1}{17}\begin{pmatrix} 7 & -3 \\ 1 & X \end{pmatrix}$, which simplifies to $\begin{pmatrix} \frac{7}{17} & -\frac{3}{17} \\ \frac{1}{17} & 2 \end{pmatrix}$, not just 17.`</think>`
`<answer>`2`</answer>`
Reward: 0.25

---

# I LLM USAGE STATEMENT

We used a large language model (OpenAI ChatGPT) as a writing assistant during the preparation of this paper. Its role was limited to language editing, including improving readability, refining phrasing, and restructuring paragraphs for clarity and conciseness. All research ideas, theoretical analyses, experimental designs, implementations, and results were conceived and carried out solely by the authors. The model did not generate novel research content, proofs, or experimental results. The authors take full responsibility for the accuracy and integrity of the paper.

