# OpenReview forum: "Unary Feedback as Observation: Incentivizing Self-Reflection in Large Language Models via Multi-Turn RL"
_ICLR.cc/2026/Conference — Submitted to ICLR 2026_

### Official Review · Reviewer_1iZH · 2025-10-30

**Soundness:** 2
**Presentation:** 2
**Contribution:** 1
**Rating:** 4
**Confidence:** 4

**Summary:**

This paper investigates Large Language Model (LLM) reasoning and Reinforcement Learning (RL) fine-tuning in a multi-turn interactive setting. It addresses a key limitation of traditional RL with Verifiable Reward (RLVR), which uses a single-turn paradigm. This traditional approach often causes models to fail to explore alternative reasoning paths or reflect on prior mistakes, leading to repetitive and unadapted responses to feedback.
To address this, the authors propose Unary Feedback as Observation (UFO), a framework that conditions policy updates on minimal unary feedback (e.g., Let’s try again) following incorrect answers.
Experiments demonstrate that UFO achieves superior performance and exhibits cross-domain generalization ability.

**Strengths:**

The work successfully identifies the importance of multi-turn interaction for LLM agents.
It shows that RL training under a multi-turn setting effectively incentivizes diversity in reasoning paths, thereby improving final performance.

**Weaknesses:**

1. The motivation for considering multi-turn interaction to encourage exploration and revision is sound. However, the paper's current setting seems limited to minimal unary feedback. It's unclear how this approach generalizes or applies to more natural multi-turn dense feedback scenarios (e.g., detailed human labels or code debugging feedback), as alluded to in paragraph 136. This limits the scope of the proposed method's applicability.
2. The algorithmic novelty appears limited. The approach essentially trains with multi-step environmental feedback without a clearly defined adaptive algorithm design. The distinction between the proposed method and previous single-step PPO/GRPO algorithms that also trained with repeated answer generation is not sufficiently clear
3. Regarding the mathematical reasoning problem, specifically:

	How is the end of a single generation step determined in this multi-turn setting?

	What is the fundamental difference between UFO and previous PPO/GRPO work, beyond the multiple interactions (turns) and the use of more data examples?

**Questions:**

1.	I am confused about Figure 1. What is the testing task used to generate this figure, and what is the specific definition of the effective answer ratio metric? A clearer explanation of why traditional RL training methods like PPO and GRPO show a performance drop compared to the original model is needed, perhaps with an earlier  and detailed explanation of the experiments would be better.

2.	In Equation 1, the formulation for the LLM generating repeated responses is introduced. To accurately model this repetition, shouldn't the LLM's input be modified to include its previous outputs? Specifically, should the formulation be adapted to something like $q(y|x) \times q(y|x, y_{prev})$, where $y_{prev}$ represents the previous response?

---

> ### Author Response · Authors · 2025-11-29
> **Response to Reviewer 1iZH**
>
> We thank the reviewer for the thoughtful and detailed comments. We respond to each concern in turn.
>
> **W1. How does this approach generalize to natural multi-turn dense feedback?**
>
> **Response.** We respectfully clarify that this concern is based on a different intended scope. Our work deliberately focuses on the weakest and most universally available feedback signal: unary feedback (“Try again”).
>
> This design choice is not a limitation; it is precisely what maximizes generality:
>
> - Any user can express that an answer is unsatisfactory, regardless of domain expertise.
> - In contrast, dense feedback (e.g., detailed comments, debugging hints) requires significantly more effort and expertise and is not universally available.
>
> If a model can learn effectively under such minimal supervision, then incorporating richer multi-turn feedback becomes easier, not harder. Unary feedback is, in this sense, a lower bound on feedback richness: natural dense feedback is a strictly stronger special case.
>
> Thus, using the simplest feedback signal broadens applicability rather than restricting it. It ensures that UFO is compatible with virtually any interaction scenario, while richer feedback can be layered on top in future extensions.
>
> **W2. Limited algorithmic novelty; unclear distinction from PPO/GRPO methods that also train with repeated answer generation.**
>
> **Response.** To the best of our knowledge, prior PPO/GRPO works operate in a single-turn or single-trajectory setting with richer reward structures (e.g., correctness signals, heuristic scores, or learned reward models). They do not address multi-turn RL driven purely by unary feedback on datasets that are intrinsically single-turn.
>
> Our contributions are:
>
> 1. **Multi-turn adaptation with unary feedback.** We adapt single-turn RL to a multi-turn setting where the only feedback is unary (“try again”). This is conceptually different from single-turn PPO/GRPO, which optimizes a single answer per prompt.
> 2. **Transforming single-turn data into a multi-turn environment.** Many existing datasets (math reasoning, theorem proving, QA) contain only one question and one final answer, without step-level labels or dense rewards. Traditional RL cannot straightforwardly operate in multi-turn mode on such data. UFO introduces a mechanism that transforms these single-turn datasets into a multi-turn environment without requiring correctness labels or handcrafted reward shaping.
> 3. **Scaling multi-turn RL with minimal supervision.** Our framework shows that multi-turn RL can be made feasible at scale using only unary signals, and that this suffices to learn multi-step refinement behaviors.
>
> If there are prior PPO/GRPO works that (i) perform multi-step repeated answer generation in a multi-turn environment and (ii) use purely unary feedback in this way, we would be very grateful for specific pointers and will cite them appropriately.
>
> **W3. Clarification of technical details: end of generation and distinction from prior PPO/GRPO.**
>
> **Response.**
>
> - **How is the end of a single generation step determined?**
>   A generation step ends when the model emits its EOS token or reaches the predefined maximum generation length. This is identical to standard PPO/GRPO setups.
>
> - **What fundamentally distinguishes UFO from previous PPO/GRPO approaches?**
>   Beyond the use of multiple interactions (turns), the key distinction is that UFO enables multi-turn interaction on single-turn datasets using only unary feedback. Traditional RL pipelines assume access to correctness labels or explicit rewards for each answer. In contrast, UFO requires only minimal “satisfactory / unsatisfactory” signals and uses these to progressively refine answers across turns. This makes it possible to train multi-turn refinement skills at scale without additional annotation.
>
> **Q1. What is the testing task in Figure 1? What is the definition of “effective answer ratio”? Why do PPO and GRPO show an apparent “drop” compared to the original model?**
>
> **Response.** Figure 1 is an illustrative example meant to visualize the multi-turn process, while the actual testing tasks and benchmarks are fully detailed in the Experiments section.
>
> The “effective answer ratio” is defined on lines 153–154 as the fraction of distinct attempts produced by a model across multiple turns. Given answers $y_1, \dots, y_T$, we count how many are unique and divide by $T$. This metric captures exploration diversity, not accuracy.
>
> The “drop” observed for PPO and GRPO reflects reduced exploration diversity after RL, not reduced capability. As discussed in Section 2.3, standard RL training reduces policy entropy, which naturally makes the model more deterministic and more likely to repeat previous answers. This leads to a lower effective answer ratio even when accuracy has improved. This behavior aligns with concurrent findings (e.g., Yue et al., “Does Reinforcement Learning Really Incentivize Reasoning Capacity in LLMs Beyond the Base Model?”).

---

> > ### Author Response · Authors · 2025-11-29
> >
> > **Q2. Why is Equation (1) written as a simplified, memoryless abstraction, and how does this relate to real multi-turn conditioning?**
> >
> > **Response.** Equation (1) is intentionally written as a simplified, memoryless abstraction to isolate the core mechanism we want to analyze: how entropy reduction in single-turn RL increases the probability of answer repetition.
> >
> > The central quantity, $\sum_y q(y \mid x)^2$, is tightly coupled to the entropy $H(q)$. Lower entropy implies higher concentration on a few modes, which increases the probability that two independent samples coincide, i.e., that the model repeats an answer. This abstraction allows for a clean theoretical treatment that directly links entropy collapse to repetition.
> >
> > In practice, real LLMs in multi-turn settings condition on previous turns, so the true distribution is closer to $q(y_t \mid x, y_{<t})$. Incorporating this full history into the theoretical derivation would significantly complicate the analysis while not changing the key qualitative conclusion: lower-entropy base policies are more prone to repetition.
> >
> > We will clarify in the paper that Equation (1) is a deliberate abstraction used solely for theoretical analysis of entropy and repetition, and that the multi-turn conditioning in real systems is richer than this simplified model.

---

### Official Review · Reviewer_ZTSc · 2025-11-01

**Soundness:** 2
**Presentation:** 2
**Contribution:** 2
**Rating:** 4
**Confidence:** 4

**Summary:**

This paper proposes UFO (Unary Feedback as Observation), a multi-turn reinforcement learning framework that enables LLMs to learn self-reflective reasoning from minimal feedback (e.g., “Try again”) on static single-turn datasets. UFO treats past attempts and unary feedback as part of the observation state, trains with PPO, and introduces reward decay and repetition penalties.

**Strengths:**

1. A simple and effective method to push LLMs learn self-reflective reasoning.

2. The reward decay and repetition penalties are introduced to encourage minimality (solving quickly) and diversity (avoiding repeated errors).

**Weaknesses:**

- The paper contains several errors that need correction. For example, Figure 7 is not cited in the paper, and the term “Multiturn” on line 462 should be hyphenated as “Multi-turn.”
- Since the authors provide the model with a prompt that includes prior attempts and feedback, it is difficult to disentangle whether performance gains come from true multi-turn interaction or simply from a richer prompt signal.
- **An Important question:** The experimental results suggest that models trained with multi-turn RL appear to require multi-turn evaluation to achieve improved performance. Does this imply that the model’s intrinsic reasoning capability—under single-turn evaluation—has not actually improved, and that richer prompt signals are still necessary to activate its parametric knowledge?
- Section 2.3 should introduce and cover the content of Figure 1.
- In Figure 2, what exactly is meant by "effective answer ratio"? Does it refer to answers that are both correct and derived via diverse reasoning paths? Which models were used in this analysis? How was the metric computed? Does the drop after RL imply that the model’s overall capability degrades?
- In Table 1, what is the "hotQA" dataset used for Qwen-3B? Why was this experiment conducted only on Qwen-3B and not on other models?
- Why did the authors choose PPO as the base RL algorithm instead of alternatives like GRPO or DAPO?
- As shown in Table 1, models trained on math datasets (e.g., MMQ-Math) generalize well to QA tasks, but models trained on HotpotQA perform poorly on math tasks. What explains this asymmetry?
- Why are hyperparameters (e.g., T_max, N) kept identical across models of different sizes? Shouldn’t larger models potentially benefit from different settings (e.g., more rollouts or longer interaction horizons)?

**Questions:**

Please see weaknesses.

---

> ### Author Response · Authors · 2025-11-29
> **Response to ZTSc**
>
> We thank the reviewer for the helpful comments and careful reading of the paper. We respond to each point below.
>
> **W1. The paper contains several errors (missing Figure 7 citation; “Multiturn” typo).**
>
> **Response.** Thanks for pointing these out. These are purely editorial issues and do not affect the technical content or conclusions of the paper. In the revised version, we will (i) add the missing citation to Figure 7 in Section 4 and (ii) correct all occurrences of “Multiturn” to “Multi-turn”, and have carefully re-read the manuscript to avoid similar minor issues.
>
> **W2. How to disentangle whether improvements come from true multi-turn interaction versus richer prompt signals?**
>
> **Response.** We agree this is a fundamental question. Our view is that in real-world LLM use, multi-turn interaction and richer prompt signals are inherently coupled. Whenever users interact across multiple turns, they inevitably provide some additional context beyond a minimal “Try again” signal, so fully disentangling the two is neither realistic nor necessarily desirable.
>
> Our contribution is to study the minimal multi-turn setting where the only additional signal is unary feedback (“try again”). This isolates the weakest plausible real-world feedback and tests whether improvement still arises when the prompt is kept as weak and uninformative as possible. Figure 7 directly supports this: under identical multi-turn setups, adding unary feedback on top of “no feedback” yields clear gains attributable solely to the feedback signal.
>
> At the same time, multi-turn structure is essential. Without multiple turns, RL from feedback cannot operate, since there is no opportunity for the model to revise its answer based on the signal. In short, both ingredients are necessary and complementary:
>
> - Multi-turn structure provides the mechanism (opportunities for revision).
> - Unary feedback provides the learning signal (when to revise).
>
> Our experiments show that even under the weakest plausible feedback, their combination yields meaningful improvements.
>
> **W3. Does requiring multi-turn evaluation mean single-turn reasoning ability has not improved? Does the model need richer prompts to “activate” its knowledge?**
>
> **Response.** We respectfully clarify that this interpretation is **not accurate**. Our results show that multi-turn RL improves both single-turn and multi-turn performance.
>
> Even under single-turn evaluation (Succ@1), models trained with multi-turn RL substantially outperform the base model. For example, in our math benchmarks, base models achieve around 20% Succ@1, whereas multi-turn-trained models reach around 80% Succ@1. This improvement is measured in a purely single-turn setting, without any multi-turn context or richer prompts, indicating a genuine strengthening of single-turn reasoning.
>
> What multi-turn RL additionally provides is a strong advantage in multi-turn evaluation. Unlike single-turn baselines, multi-turn-trained models do not saturate early; as shown in Fig. 6, they continue improving across turns and reach a higher ceiling.
>
> Thus, multi-turn RL:
>
> - Improves single-turn reasoning, as seen in Succ@1.
> - Further improves multi-turn performance by enabling sustained gains across turns.
>
> Multi-turn evaluation reveals additional benefits but is not required to observe the core improvement.
>
> **W4. Section 2.3 should introduce and cover the content of Figure 1.**
>
> **Response.** Thank you for pointing this out. Currently, Figure 1 is introduced in Section 2.2 (line 152), where we discuss the empirical multi-turn phenomena. Section 2.3 then builds a theoretical analysis on top of these phenomena and therefore does not need to re-introduce the figure.
>
> In the revision, we will make this structure explicit by (1) adding a forward reference in Section 2.2 explaining that Section 2.3 provides theoretical analysis of the behaviors illustrated in Figure 1, and (2) clarifying in Section 2.3 that it builds directly on Figure 1.

---

> > ### Author Response · Authors · 2025-11-29
> >
> > **W6. What is “hotQA” in Table 1, and why only evaluated on Qwen-3B?**
> >
> > **Response.** “hotQA” refers to HotpotQA. We will expand this abbreviation to “HotpotQA” in Table 1 and the surrounding text.
> >
> > The HotpotQA experiment is an additional ablation to test UFO on a non-math QA dataset. Because this is not part of the main nine-task comparison, we initially evaluated it only on Qwen2.5-3B-Instruct to keep the compute manageable.
> >
> > To address the reviewer’s concern, we additionally ran HotpotQA experiments on Qwen2.5-1.5B-Instruct and Llama3.2-3B-Instruct. The results below confirm the same conclusion: multi-turn RL improves performance across both math and QA tasks.
> >
> > ---
> > **Dataset abbreviations:**
> > MM = MMQ-Math, TH = TheoremQA, GS = GSM8k, GQ = GPQA, MS = MMLU-STEM, HP = HotpotQA, CQ = ConcurrentQA, MU = MMLU, MP = MMLU-Pro
> >
> > ---
> >
> > **Qwen2.5-1.5B-Instruct**
> >
> > | Turn |  MM | TH| GS| GQ | MS | HP | CQ | MU | MP|
> > |---|--|--|--|--|--|---|--|----|---|
> > | Base | 10.9% | 11.7%| 26.6%  | 21.9%| 62.5%| 2.4%| 3.1%| 52.3%| 35.2%|
> > | 1| 31.6%| 18.0%| 34.0%| 33.2%  | 62.9%  | 29.7% | 6.6%| 35.9%| 30.1%|
> > | 5| 36.3%| 18.0%| 50.0%| 29.3%  | 67.2%  | 32.8% | 8.2%| 54.3%| 32.8%|
> >
> > **Llama3.2-3B-Instruct**
> >
> > | Turn | MM | TH | GS | GQ | MS | HP | CQ | MU | MP |
> > |-|-|--|--|--|--|--|---|---|--|
> > | Base | 50.8%| 20.3%| 48.4%| 47.7%| 77.3%| 29.7%| 6.0%| 65.6%| 49.2%  |
> > | 1    | 25.0%  | 19.1%  | 21.5%  | 65.2%  | 83.6%  | 43.8%  | 10.2% | 81.6%  | 57.8%  |
> > | 5    | 46.5%  | 23.0%  | 33.6%  | 63.3%  | 79.9%  | 49.0%  | 11.3% | 83.2%  | 54.7%  |
> >
> > The overall conclusions of the paper remain unchanged.
> >
> > **W7. Why PPO instead of GRPO or DAPO?**
> >
> > **Response.** Our method, UFO, is complementary to the underlying RL algorithm and does not rely on PPO specifically. To verify this, we ran additional experiments substituting PPO with GRPO. Below we report GRPO-based results, showing that 5-turn UFO improves performance on top of GRPO as well, consistent with the PPO setting.
> >
> > **Qwen2.5-1.5B-Instruct (GRPO)**
> >
> > | Turn  | MM     | TH     | GS     | GQ     | MS     | HP     | CQ    | MU     | MP     |
> > |--|-|--|--|--|--|--|--|--|--|
> > | Base  | 10.9%  | 11.7%  | 26.6%  | 21.9%  | 62.5%  | 2.3%   | 3.1%  | 52.3%  | 35.2%  |
> > | 1| 75.0%  | 21.9%  | 85.9%  | 18.0%  | 57.0%  | 25.0%  | 7.0%  | 52.3%  | 28.9%  |
> > | 5| 81.2%  | 26.6%  | 87.5%  | 20.3%  | 61.7%  | 26.6%  | 7.0%  | 56.3%  | 38.3%  |
> >
> > **Qwen2.5-3B-Instruct (GRPO)**
> >
> > |Turn|MM|TH| GS| GQ| MS| HP| CQ| MU | MP|
> > |---|--|---|---|---|---|---|--|---|--|
> > | Base  | 52.3%  | 28.1%  | 68.0%| 51.6%| 75.8%  | 7.8%| 3.9%| 75.0%  | 42.2%  |
> > | 1| 75.8%  | 35.2%  | 89.1%| 32.0%| 77.3% | 24.2%| 14.1% | 67.2%  | 46.1%  |
> > | 5| 88.3%  | 35.2%  | 91.4%| 34.4%| 78.1%| 32.0%| 8.6%  | 76.6%  | 46.1%  |
> >
> > Thus, the choice of PPO is not essential to our conclusions; UFO can be layered on top of PPO, GRPO, or other RL algorithms..
> >
> > **Q1. Why does training on math generalize to QA, but training on QA does not generalize to math?**
> >
> > **Response.** We also found this asymmetry striking. Our interpretation is that math tasks require more structured and constrained forms of reasoning: variable extraction, step-by-step decomposition, symbolic manipulation, logical consistency, and verification. These skills form reusable subroutines that transfer naturally to QA tasks that require reasoning over text.
> >
> > By contrast, QA tasks rely more on retrieval, evidence aggregation, and summarization skills. These are valuable but do not fully cover the symbolic and verification-heavy behaviors required in mathematics. In this sense, the skill set learned from math is strictly richer and more structured, so transfer from math → QA is effective, whereas QA → math is limited.
> >
> > A deeper investigation of this asymmetry is an interesting direction for future work, but it is orthogonal to our main claims.
> >
> > **Q2. Why are hyperparameters kept identical? Will larger models potentially benefit from different settings?**
> >
> > **Response.** We intentionally keep all hyperparameters fixed across model sizes to ensure an apples-to-apples comparison. Our goal is to isolate the effect of unary-feedback multi-turn training, not to optimize each model individually.
> >
> > If we tuned hyperparameters separately for each model (e.g., different rollout horizons or learning rates), it would be difficult to attribute observed differences to UFO versus hyperparameter changes. By using a unified configuration, we can cleanly attribute improvement to the multi-turn unary-feedback mechanism.
> >
> > It is very plausible that larger models could benefit from larger N or longer horizons. Studying such scaling laws is an interesting direction, but it focuses on finding optimal per-model settings. Our current goal is more basic: to show that, under the same training budget and with the weakest possible feedback signal, multi-turn RL consistently improves performance across model sizes. Model-specific tuning can be explored in future work without affecting the validity of the current conclusions.

---

### Official Review · Reviewer_xBUi · 2025-11-03

**Soundness:** 3
**Presentation:** 2
**Contribution:** 2
**Rating:** 4
**Confidence:** 3

**Summary:**

This paper identifies a critical limitation in current reinforcement learning with verifiable reward (RLVR) methods for large language models (LLMs): they are trained under a single-turn paradigm, which suppresses adaptive and self-reflective reasoning in multi-turn interactions. To address this, the authors propose Unary Feedback as Observation (UFO) — a simple yet effective framework that enables multi-turn reinforcement learning by conditioning policy updates on minimal unary feedback such as “Try again.” Experiments across multiple LLMs and nine benchmarks show that UFO improves multi-turn reasoning success

**Strengths:**

1.	Demonstrates that self-reflective reasoning transfers across domains and architectures.
2.	Introduces reward-shaping principles (minimality/diversity) that improve both efficiency and adaptability.

**Weaknesses:**

1.	Unary feedback (“Try again”) is idealized; real human feedback can be more ambiguous or inconsistent.
2.	The sensitivity to decay factor γ and repetition penalty λ could be analyzed more deeply.

**Questions:**

1.	Could this unary feedback mechanism be extended to graded feedback (e.g., “close,” “partially correct”)?
2.	What are the computational costs compared to single-turn RL — is training efficiency affected?

---

> ### Author Response · Authors · 2025-11-29
> **Response to Reviewer xBUi**
>
> We thank the reviewer for the detailed and constructive comments. We address each point below.
>
> **W1. Unary feedback is idealized; real human feedback can be ambiguous or inconsistent.**
>
> **Response.** Our design explicitly acknowledges that real human feedback can be precise, ambiguous, or inconsistent. Precisely because of this, we deliberately place ourselves in an extremely weak-feedback regime: the user only needs to provide a unary “try again” signal, without specifying where the solution is wrong, which step fails, or how it should be fixed. A central message of our work is that even under such extremely low-information, noisy feedback, multi-turn RL still works and yields substantial gains.
>
> In realistic systems, large-scale, stable, high-quality human annotations are difficult to obtain, especially for multi-turn RL. To better reflect this constraint, we use GPT-4o to generate quasi-human feedback and **intentionally** allow this feedback to be fuzzy and inconsistent, rather than assuming a perfect oracle that always assigns accurate scores. Our experiments (Fig. 8) directly test robustness under such imperfect supervision: even when the signal is vague and noisy, UFO continues to deliver stable and significant improvements.
>
> Below we report performance under ambiguous GPT-4o feedback, further illustrating that UFO remains effective under weak and noisy feedback conditions:
>
> **Dataset abbreviations:**
> MM=MMQ-Math, TH=TheoremQA, GS=GSM8k, GQ=GPQA, MS=MMLU-STEM, HP=HotpotQA, CQ=ConcurrentQA, MU=MMLU, MP=MMLU-Pro
>
> **Performance under ambiguous feedback (Qwen2.5-3B)**
>
> |Model|MM|TH|GS|GQ|MS| HP | CQ| MU| MP |
> |-------|-------|------|---|---|-----|----|---|--|---|
> | Qwen2.5-3B-Instruct | 35.6%    | 18.2%     | 50.2% | 28.9% | 74.4%   | 5.0%     | 2.0%| 55.8% | 47.7%  |
> | Qwen2.5-3B-UFO-1turn | 84.0%    | 24.2%     | 91.0% | 21.1% | 76.2%   | 26.6%    | 12.1% | 46.9% | 41.8%  |
> | Qwen2.5-3B-UFO | 85.5% | 27.7% | 94.1% | 34.4% | 78.9%   | 37.5%| 16.8% | 59.4% | 48.4%  |
>
> Even though the quasi-human feedback itself is ambiguous and potentially inconsistent, multi-turn UFO still achieves large improvements over both the base model and the 1-turn variant. This shows that our method does not rely on idealized, high-quality human feedback; instead, it remains effective even when the feedback is extremely weak and noisy.
>
> **W2. Sensitivity to decay factor γ and repetition penalty λ could be analyzed more deeply.**
>
> **Response.** We agree that understanding sensitivity is important. Below we provide a full sensitivity study for γ and λ. Overall, γ = 0.5 and λ = 1.0 achieve the best performance across the choices we explored, which is exactly what we adopt in our paper.
>
> **γ – Success rate**
>
> | γ   | MM | TH | GS | GQ | MS | HP | CQ | MU| MP |
> |-------|---|----|---|------|-----|----------|-----|----|---|
> | 0.0   | 88.7%| 31.8% | 93.3% | 46.4% | 84.0%   | 37.0%    | 16.7% | 71.5% | 56.5%  |
> | 0.5   | 92.9% | 35.8% | 95.6% | 58.7% | 91.6%   | 37.3%    | 16.2% | 85.4% | 62.0%  |
> | 1.0   | 89.5% | 34.9% | 91.0% | 57.8% | 89.0%   | 40.3%    | 15.2% | 81.3% | 60.4%  |
>
> **γ – Average turns**
>
> | γ   | MM     | TH     | GS     | GQ     | MS     | HP     | CQ     | MU     | MP     |
> |-------|----|------|---|------|---|----------|-----|------|---|
> | 0.0   | 1.3 | 2.2 | 1.1 | 2.3 | 1.5 | 2.4 | 4.2 | 1.7 | 2.0 |
> | 0.5   | 1.3 | 2.8 | 1.3 | 3.1 | 1.7 | 3.7 | 4.5 | 2.1 | 2.5 |
> | 1.0   | 1.6 | 3.0 | 1.6 | 3.1 | 1.8 | 3.6 | 4.5 | 2.2 | 2.7 |
>
> **λ – Success rate**
>
> | λ   | MM     | TH     | GS     | GQ     | MS     | HP     | CQ     | MU     | MP     |
> |--------|--------|-----------|-------|------|-----------|----------|--------------|------|----|
> | 0.0    | 90.3%    | 31.6%     | 93.5% | 44.7% | 85.4%   | 32.5%    | 14.5%        | 77.8% | 55.7%  |
> | 1.0    | 92.9%    | 35.8%     | 95.6% | 58.7% | 91.6%   | 37.3%    | 16.2%        | 85.4% | 62.0%  |
> | 2.0    | 90.2%    | 32.9%     | 95.0% | 51.5% | 91.5%   | 38.5%    | 15.8%        | 82.1% | 61.4%  |
>
> **λ – Average turns**
>
> | λ   | MM     | TH     | GS     | GQ     | MS     | HP     | CQ     | MU     | MP     |
> |----|----|---|-------|------|----|--|----|---|---|
> | 0.0    | 1.4 | 3.0 | 1.3 | 3.3 | 1.8 | 3.8 | 4.5 | 2.1 | 2.7 |
> | 1.0    | 1.3 | 2.8 | 1.3 | 3.1 | 1.7 | 3.7 | 4.5 | 2.1 | 2.5 |
> | 2.0    | 1.2 | 2.1 | 1.1 | 2.7 | 1.6 | 3.5 | 4.5 | 2.0 | 2.3 |
>
> **Additional clarification.**
> When γ = 0, only first-turn success is rewarded, effectively reducing training to a 1-turn setting. When γ = 1, later turns are rewarded equally, encouraging longer interactions. The parameter λ controls the strength of the repetition penalty: larger λ suppresses repeated answers more strongly, while smaller λ allows more repetition.
>
> We are happy to include these hyperparameter studies and extend them further in the camera-ready version if the reviewer has specific suggestions.

---

> > ### Author Response · Authors · 2025-11-29
> >
> > **Q1. Could unary feedback be extended to graded feedback?**
> >
> > **Response.** In principle, unary feedback can be extended to richer signals such as “close” or “partially correct,” but this is beyond the scope of our current work.
> >
> > Crucially, graded signals assume the user can judge how an answer is wrong or which steps are partially correct, which substantially increases the expertise and cognitive burden. In contrast, our unary feedback only requires the user to know that “the answer is unsatisfactory.” This minimal requirement is key to practical deployment.
> >
> > Moreover, for our QA and math domains, partial correctness is often not reliably quantifiable: different users may disagree on what fraction of a multi-step proof is “correct enough.” We therefore focus on unary feedback as the most general, no-brainer signal with minimal assumptions. Extending UFO to richer feedback types is a natural direction for future work.
> >
> > **Q2. What are the computational costs vs. single-turn RL?**
> >
> > **Response.** We agree that compute tradeoffs matter. While 5-round training requires more compute per step than 1-round training, the learning dynamics justify this cost. As shown in our training curves (e.g., Fig. 4), both models start with similar success rates, but the 1-round curve quickly saturates around ~0.80, whereas the 5-round curve continues to improve and ultimately reaches a higher plateau of ~0.92. This demonstrates that multi-turn rollouts extract more supervision from each trajectory and convert the same dataset into more effective improvements. In other words, multi-turn RL provides strictly better sample efficiency and a higher performance ceiling, which is desirable when the goal is to maximize reasoning capability.
> >
> > A more precise wall-clock analysis shows that the total training time over 200 steps grows much less than the naïve 5× expectation. For Qwen2.5-3B-Instruct on MMQ-Math, 5-round training takes only ~1.8× the wall-clock time of 1-round training over the same 200 steps; for Llama-3.2-3B-Instruct on HotpotQA, it is ~3.5×. This difference stems from the fact that higher-accuracy models terminate earlier, reducing the number of turns. In practice, even this ~1.8–3.5× increase in total training time yields a markedly higher final success rate (e.g., ~0.80 → ~0.92), which we view as a highly favorable tradeoff.

---

### Author Response · Authors · 2025-11-29

General response

## Summary
We thank the AC and reviewers for their careful reading and constructive feedback. Below we summarize our thesis, key empirical findings, and how our new experiments and clarifications address the main concerns across the three reviews.

## Thesis
Our goal is to study multi-turn RL under the **weakest realistic supervision signal**: unary “try again” feedback that does not require any task-specific expertise or correctness labels. We show that even under this minimal and noisy signal, multi-turn RL can significantly improve both single-turn and multi-turn reasoning, and that our UFO framework can turn single-turn datasets into effective multi-turn training environments.

## Key results

1. **Multi-turn RL under minimal unary feedback works.**
   Our multi-turn RL with unary “try again” feedback, implemented using ambiguous and sometimes inconsistent GPT-4o quasi-human feedback, **raises multi-turn success from about 35% to about 85%** on MMQ-Math benchmark. We also tested on 9 different datasets across 4 domains, and the improvement is universal. These results show that the method does not need rich step-level labels or dense rewards to obtain large gains.

2. **Compute–performance tradeoff is favorable and quantified.**
   We profile Qwen2.5-3B-Instruct (MMQ-Math) and Llama-3.2-3B-Instruct (HotpotQA) and **measure that 5-round training increases time per step by only about 1.8–3.5×**, much smaller than the naive 5× factor. These measurements show that multi-turn UFO achieves **stricttly better sample efficiency and a higher final success rate** at a moderate and well-quantified compute cost.

3. **UFO is a general multi-turn RL framework, not tied to a specific RL algorithm.**
   UFO converts intrinsically single-turn datasets (math, theorem proving, QA) into multi-turn environments using **only unary feedback**, and scales in this setting **without correctness labels or handcrafted reward shaping**. Our experiments show consistent gains when we apply UFO on top of both PPO and GRPO and across multiple model sizes, which demonstrates that the framework is complementary to existing RL algorithms.

4. **Extensive new experiments strengthen empirical rigor.**
   We add a full sensitivity analysis for the decay factor **γ** and repetition penalty **λ**, additional HotpotQA and QA results on more models (Qwen2.5-1.5B-Instruct and Llama-3.2-3B-Instruct), extra GRPO-based runs, and clarifications of metrics such as the **effective answer ratio**. These results all support the same qualitative conclusions as the original submission.

## Rebuttal summary

1. **Realism and scope of unary feedback (xBUi-W1, xBUi-Q1, 1iZH-W1).**
   We clarify that our design explicitly assumes human feedback can be ambiguous, or inconsistent, and we intentionally operate in an extremely weak and noisy feedback regime. New experiments with ambiguous GPT-4o feedback show that UFO remains robust, and we argue that richer natural feedback is a strictly easier special case.

2. **Effectiveness vs. compute cost and multi-turn evaluation (xBUi-Q2, ZTSc-W2, ZTSc-W3).**
   We show that multi-turn RL improves both single-turn and multi-turn performance (Succ@1 ~20% → ~80%; multi-turn ceiling ~0.80 → ~0.92), and that 5-round training adds at most ~3.5× time per step rather than 5×. Multi-turn evaluation exposes extra gains but is not required to see improved single-turn reasoning.

3. **Novelty and relation to prior PPO/GRPO-style methods (ZTSc-W2, 1iZH-W2, 1iZH-W3).**
   We emphasize that UFO is, to our knowledge, the first framework to enable multi-turn RL on single-turn datasets using only unary feedback and no correctness labels, and to demonstrate this at scale across math and QA. We add GRPO experiments to show that UFO consistently improves performance on top of different RL algorithms.

4. **Additional analyses and clarifications (xBUi-W2, ZTSc-W4, ZTSc-W5, ZTSc-W6, ZTSc-W7, ZTSc-Q1, ZTSc-Q2, 1iZH-Q1, 1iZH-Q2).**
   - We run full γ and λ sensitivity studies and find that the chosen values (γ = 0.5, λ = 1.0) are near-optimal within our search space.
   - We define the effective answer ratio clearly, explain its role, and show why RL tends to lower this diversity metric even while it improves accuracy.
   - We add new HotpotQA and cross-model results and analyze why math training transfers well to QA but QA training transfers poorly to math.
   - We keep hyperparameters identical across model sizes to support clean, apples-to-apples comparisons and explain this design choice.
   - We clarify that Figure 1 serves as an illustrative schematic rather than an evaluation plot, and that Equation (1) is a simplified theoretical abstraction used only for analysis.


5. **Editorial fixes (ZTSc-W1).**
   We will fix the missing citation to Figure 7, correct all “Multiturn” typos to “Multi-turn,” and carefully proofread the camera-ready version for similar minor issues.

Best regards,

The Authors

---

### Meta-Review · Area_Chair_hLUR · 2026-01-08

**Summary:**

The paper proposes eliciting self revision using the minimal possible informative feedback (e.g., try again). This could be seen as a form of sequential pass at n. The authors show the advantage of the method.

**Reviewer Concerns:**

Reviewers are concerned about limited novelty, and whether a richer feedback is actually the source of performance improvement. Overall, they recommend rejection

**Reviewer Scores:**

Reviewers agrees in rejecting the paper, so it appears unlikely that most could be flipped towards an acceptance.

---

### Decision · Program_Chairs · 2026-01-26

Reject